# Lipolysis-derived fatty acids are needed for homeostatic control of sterol element-binding protein-1c driven hepatic lipogenesis

Paola Peña de la Sancha [1,2,8], Beatrix Irene Wieser [1,2,8], Silvia Schauer[1], Helga Reicher[3], Wolfgang Sattler[2,3], Rolf Breinbauer [2,4], Martina Schweiger [2,5], Margarete Lechleitner[3], Saša Frank[3], Rudolf Zechner[2,5], Dagmar Kratky[2,3], Peter John Espenshade [6], Gerald Hoefler[1,2] & Paul Willibald Vesely [1,2,7] ✉

Sterol Regulatory Element-Binding Protein-1c (SREBP-1c) is translated as an inactive precursor (P-SREBP-1c) postprandially. Low levels of unsaturated fatty acids (uFAs) and high insulin promote its proteolytic activation, yielding N-SREBP-1c that drives fatty acid (FA) biosynthesis. During fasting, however, lipogenesis is low, and adipose tissue lipolysis supplies the organism with FAs. Adipose Triglyceride Lipase (ATGL) is the rate-limiting enzyme for adipose tissue lipolysis, and it preferentially releases uFAs. Therefore, we hypothesized that adipose ATGL-derived uFAs suppress P-SREBP-1c activation in the liver. In this study, we show that (I) N-SREBP-1c is transiently higher in livers of fasted and refed adipose specific *Atgl* knockout mice than in control livers. (II) This effect is reversed by injection of uFAs. (III) uFAs inhibit endoplasmic reticulum to Golgi-apparatus transport of SREBP Cleavage-Activating Protein (SCAP) in hepatocytes, which is essential for SREBP activation. Our findings demonstrate that adipose tissue ATGL derived uFAs attenuate P-SREBP-1c activation in the liver mainly after refeeding. We propose that this ATGL/SREBP-1c axis adds an additional layer of coordination between lipogenesis and lipolysis.

The *Srebf1* gene, which encodes Sterol Regulatory Element-Binding Protein-1c (SREBP-1c), is transcriptionally driven by a carbohydrate rich diet and translated as an inactive membrane-bound precursor (P)-SREBP-1c. If uFAs and/or sterols are scarce, P-SREBP-1c is transported from the endoplasmic reticulum (ER) membrane to the Golgi apparatus (Golgi) together with its chaperone SCAP (SREBP Cleavage Activating Protein). There, its N-terminal transcription-factor domain (N)-SREBP-1c, is proteolytically released from the membrane. Subsequently, N-SREBP-1c enters the nucleus, where it drives the transcriptional program for proteins involved in FA and triglyceride (TG) synthesis (lipogenesis). High insulin levels activate *Srebf1* transcription and cleavage-activation of its protein product, P-SREBP-1c[1–6]. During fasting, however, lipogenesis is low, and adipose tissue lipolysis is

strongly activated by lipolytic hormones and low plasma insulin levels. This results in FA release from adipose tissue TG stores, increased plasma FA concentrations and elevated FA uptake by the liver[7]. Adipose Triglyceride Lipase (ATGL) catalyzes the first and rate-limiting step of TG lipolysis[7–10]. The gene encoding ATGL is the Patatin-like phospholipase domain containing 2 (*Pnpla2*), also known as *Atgl*[11]. Importantly, ATGL preferentially releases uFAs from TG. ATGL-knockout animals, in turn, are defective in lipolysis and show only half the fasting plasma NEFA (non-esterified FA) levels of controls. The uFAs, palmitoleic acid (16:1), oleic acid (18:1) and linoleic acid (18:2), are even further underrepresented[12,13]. Notably, these uFAs were found to suppress the proteolytic activation of P-SREBP-1c through stabilization of the ER anchor-protein of the SCAP/SREBP

[1]Diagnostic and Research Institute of Pathology, Medical University of Graz, Graz, Austria. [2]BioTechMed-Graz, Graz, Austria. [3]Division of Molecular Biology and Biochemistry, Gottfried Schatz Research Center, Medical University of Graz, Graz, Austria. [4]Institute of Organic Chemistry, Graz University of Technology, Graz, Austria. [5]Institute of Molecular Biosciences, University of Graz, Graz, Austria. [6]Department of Cell Biology, Johns Hopkins University School of Medicine, Baltimore, USA. [7]Otto Loewi Research Center, Lung Research Group, Graz, Austria. [8]These authors contributed equally: Paola Peña de la Sancha, Beatrix Irene Wieser. ✉e-mail: Paul.vesely@medunigraz.at

complex, Insulin-Induced Gene-1 protein (INSIG-1) in vitro[14–16]. However, the tissue source of these FAs remains unknown. Therefore, we hypothesized that adipose tissue lipolysis-derived FAs may contribute to the regulation of SREBP-1c in the liver. To test this hypothesis, we used mice either lacking *Atgl* in the adipose tissue[17] or in the liver[18]. The analysis of these models suggested that FA released by ATGL in adipose tissue during fasting are transported to the liver via the circulation, where they attenuate SREBP-1c cleavage-activation.

## Results

### Adipose tissue ATGL regulates SREBP-1c in the liver
The nutrient status strongly affects SREBP-1c activation in the liver. Horton et al.[2] found that feeding a high carbohydrate/low-fat diet (HChD) after prolonged fasting strongly induced SREBP-1c[2,3]. To test the role of ATGL-mediated adipose tissue lipolysis on SREBP-1c regulation in the liver, we performed a similar experiment, albeit using adipose specific ATGL deficient (AAKO) mice (*Atgl*<sup>flox/flox</sup>, *Adipoq-Cre*) or isogenic controls (*Atgl*<sup>flox/flox</sup>). The animals were either fasted overnight or fasted overnight and refed a HChD for 3, 6, and 9 h. The mice were sacrificed at the respective time points, and their livers were fractionated to obtain microsomal membrane extracts (MM) and soluble nuclear extracts (NEX). This allowed us to analyze P- and N-SREBP-1c respectively, by western blot (WB). The signals for the full-length ER-resident P-SREBP-1c protein were stronger in refed mice compared to fasted mice (Fig. 1A), and the underlying *Srebf1* mRNA was regulated in a similar manner (Supplementary Fig. 1A). The signals of the transcriptionally active N-SREBP-1c were assessed densitometrically, normalized to protein loading, and statistically evaluated. N-SREBP-1c signals were below the detection limit in fasted and 3 h refed mice. 6 h after refeeding, N-SREBP-1c signals were significantly stronger in AAKO mice than in control mice, showing a faster induction kinetic. 9 h post-refeeding, N-SREBP-1c reached similarly high levels in both groups (Fig. 1A, B). In line, the SREBP-1c target genes *Acaca* and *Fasn*, were upregulated faster in AAKO than in control livers (Fig. 1C). The SREBP-2 target genes *Hmgcr* and *Hmgcs* showed no differential regulation between the groups (Supplementary Fig. 1B).

SREBP-1c cleavage-activation is post-translationally suppressed by uFAs[14–16]. ATGL-mediated adipose tissue lipolysis releases albumin-bound FAs into the bloodstream during fasting. This increases the plasma NEFA concentration and FA uptake by the liver, where they are stored as TGs[7,11]. Consistently, we found significantly higher plasma NEFA levels in fasted control mice compared to fasted AAKO mice (Supplementary Fig. 2A). In line with the literature, our GC/FID (gas chromatography-flame ionization detection) analysis showed that uFAs levels (16:1, 18:1, 18:2, and 20:4) were influenced much more than saturated FA (sFAs) levels (16:0, and 18:0) by the lack of adipose tissue ATGL (Fig. 1D)[12,13]. As a result, the unsaturated liver NEFA concentrations were significantly elevated during fasting in control mice compared to AAKO, while saturated sFAs were relatively similar in both groups (Fig. 1E). However, after refeeding, the plasma and liver NEFA differences were less pronounced (Supplementary Fig. 2A & B, respectively). Stored liver neutral lipids behaved similarly, as shown by ORO (Oil-red-O) neutral lipid staining of liver sections from fasted mice (Fig. 1F) and by biochemical neutral lipid measurements of fasted and refed mice (Supplementary Fig. 3A, B)[17].

P-SREBP-1c cleavage is activated by insulin in a process that requires the small ribosomal subunit protein S6 (S6) phosphorylation by its kinase, the p70S6K ribosomal subunit protein S6[19]. Mice lacking ATGL show increased insulin sensitivity and the lack of ATGL in the adipose tissue improves glucose tolerance[11,13,17,20]. However, refeeding a HChD to previously fasted AAKOs or control mice did not reveal appreciable differences in plasma insulin or glucose levels (Supplementary Fig. 4A & B). To assess tissue specific insulin sensitivity in the liver, we tested the activation of AKT using a p-Ser473 (p-AKT) antibody, and activation of the p70S6K/S6 arm of insulin receptor signaling, using p-Ser240/44 S6 and p-Ser235/36 S6 specific WB antibodies[19]. No deregulation of p-AKT was apparent. However, S6 activation was significantly increased in AAKO compared to controls at 6 h and 9 h post refeeding (Fig. 1G, H).

Collectively, our findings and the cited literature suggest that reduced availability of uFAs and enhanced tissue specific insulin sensitivity lead to increased SREBP-1c cleavage (Fig. 1A, B) and upregulated SREBP-1c target gene activation (Fig. 1C) in livers of AAKO compared to controls[11,12,19,21,22].

### Liver ATGL regulates SREBP-1c in the liver
To test if ATGL plays a direct role in liver SREBP-1c regulation by catalyzing the release of uFAs from liver TG stores, we used liver-specific ATGL deficient ALKO mice (*Atgl*<sup>flox/flox</sup>, *Alb-Cre*) and isogenic controls (*Atgl*<sup>flox/flox</sup>). The animals were subjected to the same fasting/refeeding regimen described above (Fig. 1). After sacrifice, livers were resected and fractionated to perform WBs. In both controls and ALKOs, P- SREBP-1c WB signals steadily increased after refeeding (Fig. 2A). The *Srebf1* mRNA showed a faster induction kinetic in the liver-knockout group (Supplementary Fig. 5A). The transcriptionally active N-SREBP-1c protein fragment was fully induced 6 h after HChD refeeding in ALKOs, whereas, in controls, it took 9 h to reach comparable levels (Fig. 2B). Aligned with these results, the expression of the SREBP-1c target genes, *Acaca* and *Fasn* was significantly higher in the ALKO group 6 h post-refeeding compared to controls (Fig. 2C). Conversely, the SREBP-2 target genes *Hmgcr* and *Hmgcs* showed a smaller induction at 9 h refed timepoint in ALKOs compared to controls (Supplementary Fig. 5B).

To test the potential contribution of ATGL-derived FAs to hepatic SREBP-1c regulation we again analyzed plasma- and liver- NEFAs. Fasted ALKO and control mice showed similar plasma NEFA concentrations, presumably since ATGL was lacking in the liver but not in the adipose tissue (Fig. 2D)[7,12]. After refeeding, plasma NEFA levels decreased similarly in both groups as expected (Supplementary Fig. 6A). On the other hand, several of the saturated and unsaturated liver NEFAs of fasted mice were higher in ALKOs compared to controls (Fig. 2E) up to 6 hours post-refeeding (Supplementary Fig. 6B). Similarly, ORO staining indicated increased neutral lipid content in the livers of fasted ALKOs compared to controls (Fig. 2F). In line, liver neutral lipids were higher in ALKOs throughout the time course but only statistically significant at 6 and 9 h post-refeeding, compared to controls (Supplementary Fig. 7A, B). This may be explained by reduced TG lipolysis in hepatocytes lacking ATGL[18,23]. However, how these findings may explain the faster induction of N-SREBP-1c in the ALKOs compared to controls was unclear.

To explore this further, we tested if the ALKOs showed modified plasma glucose levels or insulin sensitivity. Yet, neither plasma glucose- nor plasma-insulin levels showed any significant differences between the groups (Supplementary Fig. 8A, B). To assess tissue insulin sensitivity in the liver, we again assessed AKT activation using p-Ser473 (p-AKT) antibody and S6 activation using p-Ser240/44 S6 and p-Ser235/36 S6 antibodies. However, neither AKT nor S6 phosphorylation significantly differed between AAKO and controls 6 h and 9 h after HChD refeeding (Fig. 2G, H).

Cumulatively, these data suggested that in the presence of functional ATGL in adipose tissue, the lack of liver ATGL leads to a faster induction of N-SREBP-1c in the liver. This effect might be explained by the liver steatosis observed in ALKO mice[24,25]. To test our initial hypothesis that adipose tissue lipolysis-derived FAs may contribute to the regulation of SREBP-1c in the liver, we constructed a SREBP-1c cleavage-reporter system.

### Construction and validation of a SREBP-1c cleavage-activation reporter in vitro
To enable us to directly test if adipose tissue ATGL lipolysis derived uFAs that enter the liver through the bloodstream can suppress hepatic SREBP-1c cleavage-activation, we developed a SREBP-1c cleavage-activation reporter system. First, we cloned a triple Flag-tag in front of the human *SREBF1* cDNA (Flag-SREBP-1c) and introduced it into a constitutive promoter-driven expression vector, as described before[26]. To test the resulting pFlag-SREBP-1c construct, we used the hepatic cell line HepG2. Cells were transiently transfected with pFlag-SREBP-1c, and 48 h later, we set them up in a lipid-depleted medium (5% LPDS) to activate SREBP-1c cleavage. Subsequently, the cells were either kept under lipid-depleted conditions or supplemented with albumin-bound sFAs (16:0), or albumin-bound uFAs (18:1,

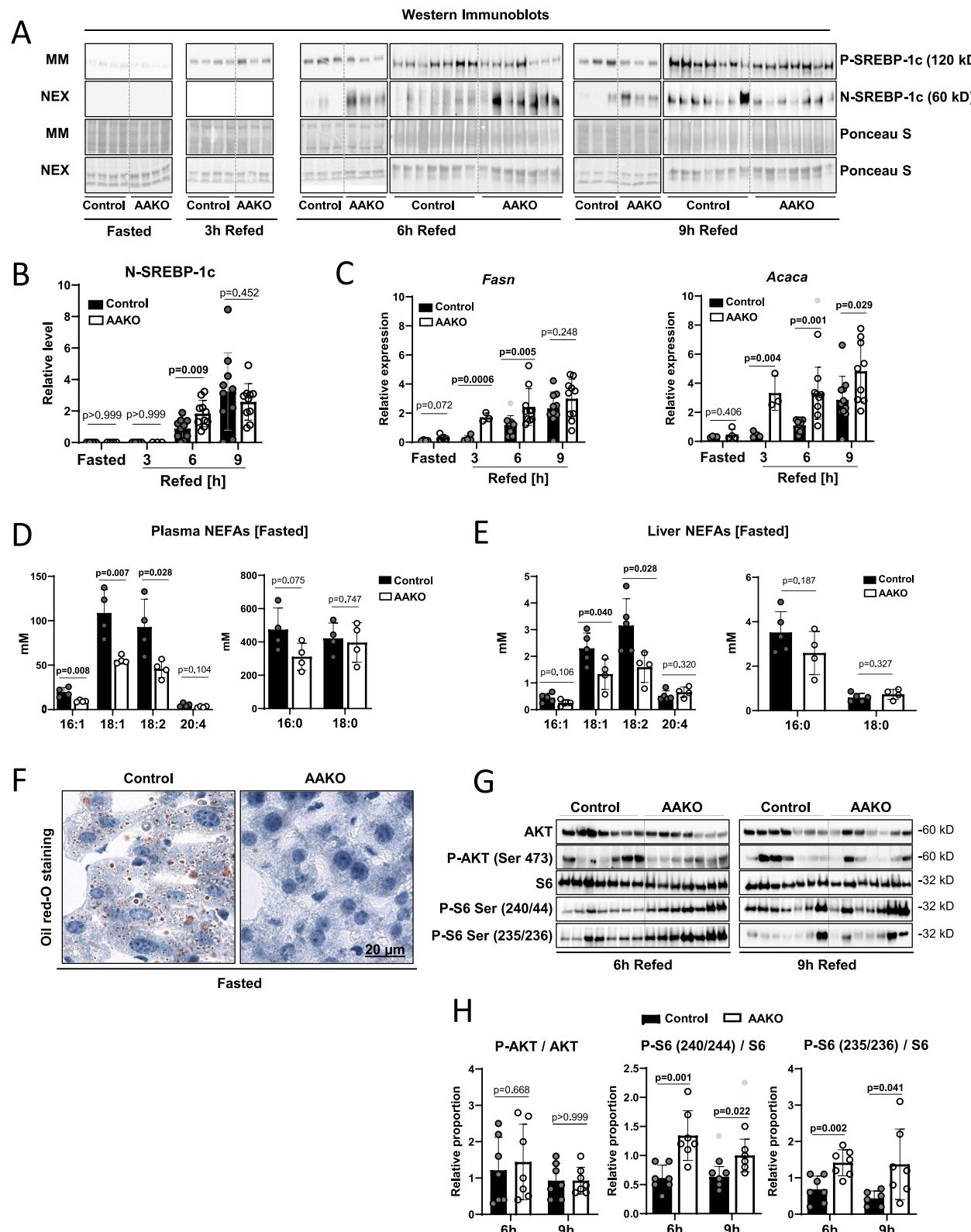

18:2, 16:1 or 20:4, respectively) for 16 h (Fig. 3A). Supplementation with 25-hydroxycholesterol (25-HC) served as cleavage-suppression control[26]. Whole-cell extracts were subjected to WB using anti-Flag antibody. P- and N-SREBP-1c were detected on the same membrane (Supplementary Fig. 9). For better visualization, however, we exposed them separately. Relative band intensities were measured densitometrically using ImageJ, NIH. Relative

levels of proteolytically cleaved N-SREBP-1c were calculated as the fraction of N-SREBP-1c/P-SREBP-1c signal intensities and are presented in the dot plot (Fig. 3B). Consistent with earlier reports, the addition of sFA (16:0) showed no effect, while treatment with uFAs (18:1, 18:2, 16:1 and 20:4) or 25-HC strongly inhibited P-SREBP-1c cleavage-activation (Fig. 3B)[14,16,26,27]. A second in vitro test using a stably pFlag-SREBP-1c transfected U2OS cell line

**Fig. 1 | Adipose tissue ATGL regulates SREBP-1c in the liver. A** Control and AAKO (adipose specific ATGL deficient) mice were fasted for 12 h overnight and subsequently either sacrificed (Fasted) or refed a HChD (high carbohydrate/low-fat diet) and sacrificed at the time-points indicated (Refed). Livers were resected, microsomal membrane fractions (MM) and soluble nuclear extracts (NEX) were prepared and subjected to Western Blot (WB) using an SREBP-1c specific antibody. Ponceau-S-stained membranes are shown as loading controls. Each lane represents a liver extract from one mouse. WBs (horizontally separated at 6 and 9 h) represent separate experiments. Band intensities were measured using ImageJ, NIH. **B** Levels of proteolytically cleaved N-SREBP-1c were normalized to protein loading and are presented in the dot plot. **C** Liver *Fasn* and *Acaca* (mRNA) levels were determined by qPCR (quantitative real-time polymerase chain reaction). **D** Plasma NEFA (non-esterified fatty acids) levels and **E** liver NEFA levels were measured using GC/FID (gas chromatography-flame ionization detection), $n = 4$–10/group. **F** Liver sections were prepared from fasted Control and AAKO mice and stained with Oil-red-O neutral lipid dye. Representative images are depicted. **G** Liver extracts were subjected to WB using the antibodies depicted. **H** Relative proportions of the respective phospho-proteins/proteins were calculated and are presented in the dot plot. $n = 7$/ group. Outliers are shown as light dots. Unpaired $t$-tests were used to compute significance levels.

(UF1c) yielded comparable results, showing that uFAs and 25-HC suppressed SREBP-1c cleavage activation, while sFAs treatment had no effect (Supplementary Fig. 10A, B). These results validated our newly established SREBP-1c cleaving-activation reporter system on a cellular level in vitro.

### Validation of SREBP-1c cleavage-activation reporter in vivo

To enable us to express the Flag-SREBP-1c construct in mouse livers, we cloned the expression cassette into an adenoviral vector using the AdEasy system, yielding the Ad-Flag-SREBP-1c vector (Fig. 4A). To test the adenoviral construct, we repeated the classic experiment described by Horton et al. in 1998, where they showed that refeeding HChD to previously fasted mice leads to SREBP-1c activation[2]. We intravenously injected $2 \times 10^9$ PFU (plaque forming units) of Ad-Flag-SREBP-1c into groups of C57/Bl6J wild type mice. Four days later, to reach stable adenoviral expression levels, the animals were either fasted overnight or fasted and refed a HChD overnight (Fig. 4A). All mice were sacrificed in the morning, livers resected and fractionated to obtain MM and NEX and subjected to WB using a Flag-tag specific antibody. Both the fasted and the HChD refed groups, showed constitutive expression of precursor (P)-Flag-SREBP-1c in the liver (Fig. 4B). This confirmed the constitutive nature of Ad-Flag-SREBP-1c expression. The nuclear (N)-Flag-SREBP-1c signals were weak under fasting and strongly induced by HChD refeeding. ImageJ was used for densitometrical band analysis, and the relative fraction of N-SREBP-1c/P-SREBP-1c in each of the livers is presented in the dot plot (Fig. 4B). This indicates that HChD refeeding after prolonged fasting significantly elevates SREBP-1c cleavage-activation[2,3,14,15]. Regulation of the SREBP-1c target genes *Fasn* and *Acaca* was in line with this finding (Fig. 4C). The endogenous *Srebf1* mRNA and the SREBP-2 targets *Hmgcr* and *Hmgcs* were also induced by HChD feeding[2] (Supplementary Fig. 11A). Unsaturated plasma NEFA levels were reduced upon HChD refeeding, however, saturated plasma NEFAs remained largely unchanged (Supplementary Fig. 12A). As expected, blood glucose and insulin levels were strongly induced postprandially (Fig. 4D). Similarly, tissue specific insulin signaling in the liver was induced by HChD refeeding (Fig. 4E, F).

Next, we tested the impact of dietary FA on Ad-Flag-SREBP-1c cleavage-activation. Thus, we again injected Ad-Flag-SREBP-1c into C57/Bl6J wild type mice. Four days later, we fed them a uFA-rich flaxseed oil diet (high 18:3 - linolenic acid content), or a sFA-rich palm oil diet (high 16:0 - palmitic acid content) for 3 days, *ad libitum* (Fig. 4A). As expected, the uFA-rich diet led to weaker Flag-SREBP-1c cleavage-activation compared to the sFA-rich diet (Fig. 4G)[14,16]. The SREBP-1c target gene *Fasn*, showed a slight upregulation in the sFA-rich diet group compared to the uFA-rich diet. Nevertheless, neither *Acaca* (Fig. 4H), endogenous *Srebf1* mRNA, nor the SREBP-2 targets *Hmgcr* and *Hmgcs* (Supplementary Fig. 11B) were significantly deregulated between the groups. Additionally, due to the high linolenic acid (18:3) content of the uFA-enriched diet, a strong increase in 18:3 plasma NEFA concentration was observed, compared to the sFA-enriched diet group. A comparably weak compensatory decrease of 16:1, 18:1, and 20:4 was also evident. Moreover, 16:0 was decreased in the uFA-rich diet group, most likely due to the sFA-rich diet's high 16:0 content (Supplementary Fig. 12B). Plasma glucose levels were relatively similar between the groups, while insulin levels were reduced by uFA-rich diet compared to sFA-rich diet (Fig. 4I). Nevertheless, tissue specific insulin signaling in the liver remained unchanged between the groups (Fig. 4J, K).

Overall, our results in Fig. 4 showed that the newly established Ad-Flag-SREBP-1c cleavage-activation reporter system reacted to HChD refeeding and to sFA- or uFA- rich diets in accordance with the literature[2,28].

### SREBP-1c cleavage-activation is suppressed by lipolysis derived uFAs

uFAs inhibit cleavage-activation of P-SREBP-1c, resulting in a reduction of nuclear N-SREBP-1c levels (Fig. 3B, and Supplementary Fig. 10)[14–16]. In the livers of fasted AAKO mice, we recorded reduced uFAs availability when compared to controls (Fig. 1E). We hypothesized that this was the cause for the enhanced N-SREBP-1c levels in AAKO livers compared to control livers during refeeding (Fig. 1A). However, the significant uFAs level differences were only evident during fasting (Fig. 1E) when the activated N-SREBP-1c was below WB detection limit (Fig. 1A) and not after refeeding (Supplementary Fig. 2A & B). Moreover, also the tissue specific insulin sensitivity in livers of AAKO was elevated compared to controls (Fig. 1G), and insulin transcriptionally and post-translationally activates SREBP-1c[29]. Therefore, it was questionable whether we observed a direct FA related effect or an indirect effect during refeeding due to elevated insulin sensitivity of the AAKOs (Fig. 1A, B, G, H).

To experimentally test if lipolysis-derived FA regulate P-SREBP-1c cleavage-activation in the liver, we used our Ad-Flag-SREBP-1c cleavage-reporter vector. P-Flag-SREBP-1c is expressed from a constitutive promoter that is not regulated by insulin. Ad-Flag-SREBP-1c was injected into control and AAKO mice, which were kept on a standard chow diet for the following 4 days, to reach stable adenoviral expression levels. Subsequently, they were fasted for 12 h overnight and sacrificed in the morning (Fig. 5A). Livers were resected, fractionated and analyzed by WB using anti-Flag specific antibody. Due to constitutive expression, relatively similar P-Flag-SREBP-1c levels were observed in all livers, regardless of the genotype. However, N-Flag-SREBP-1c levels were significantly higher in fasted AAKOs than in fasted controls (Fig. 5B). This indicated a higher P-Flag-SREBP-1c cleavage-activation rate (Fig. 5C) due to reduced uFAs availability (Fig. 1D). To test whether the exogenous supply of uFAs could rescue the enhanced cleavage-activation rate in fasted AAKOs, we repeated the experiment, except that this time, we injected the mice with bovine serum albumin (BSA) complexed oleic acid (18:1), 3 and 9 h after food withdrawal (Fig. 5A). As a result, we observed no appreciable P- nor N-Flag-SREBP-1c differences between the genotypes (Fig. 5B). This indicated that the injected uFAs suppressed P-Flag-SREBP-1c to N-Flag-SREBP-1c cleavage, even in AAKO mice with reduced adipose tissue lipolysis (Fig. 5C)[14,16,26,27]. The expression of the SREBP-1c target genes, *Fasn* and *Acaca*, reacted in a relatively similar manner as N-Flag-SREBP-1c, but *Srebf1* mRNA was unchanged (Supplementary Fig. 13A). The plasma glucose and insulin levels were lower in fasted AAKOs compared to controls. However, the oleic acid injection abolished the plasma glucose differences between the genotypes. Insulin levels were too heterogeneous after oleic acid injection to draw clear conclusions (Supplementary Fig. 13B & C). Tissue specific insulin signaling in the liver was again measured using AKT and S6 antibodies. Consistent with previous findings, AKT and S6 were weakly activated in fasted mice when compared to a positive control pool of 3 liver extracts from 6 h HChD refed control mice (Pool of 6 h refed) (Fig. 5D)[19]. Therefore, we could only quantify Ser 240/44 phosphorylation status

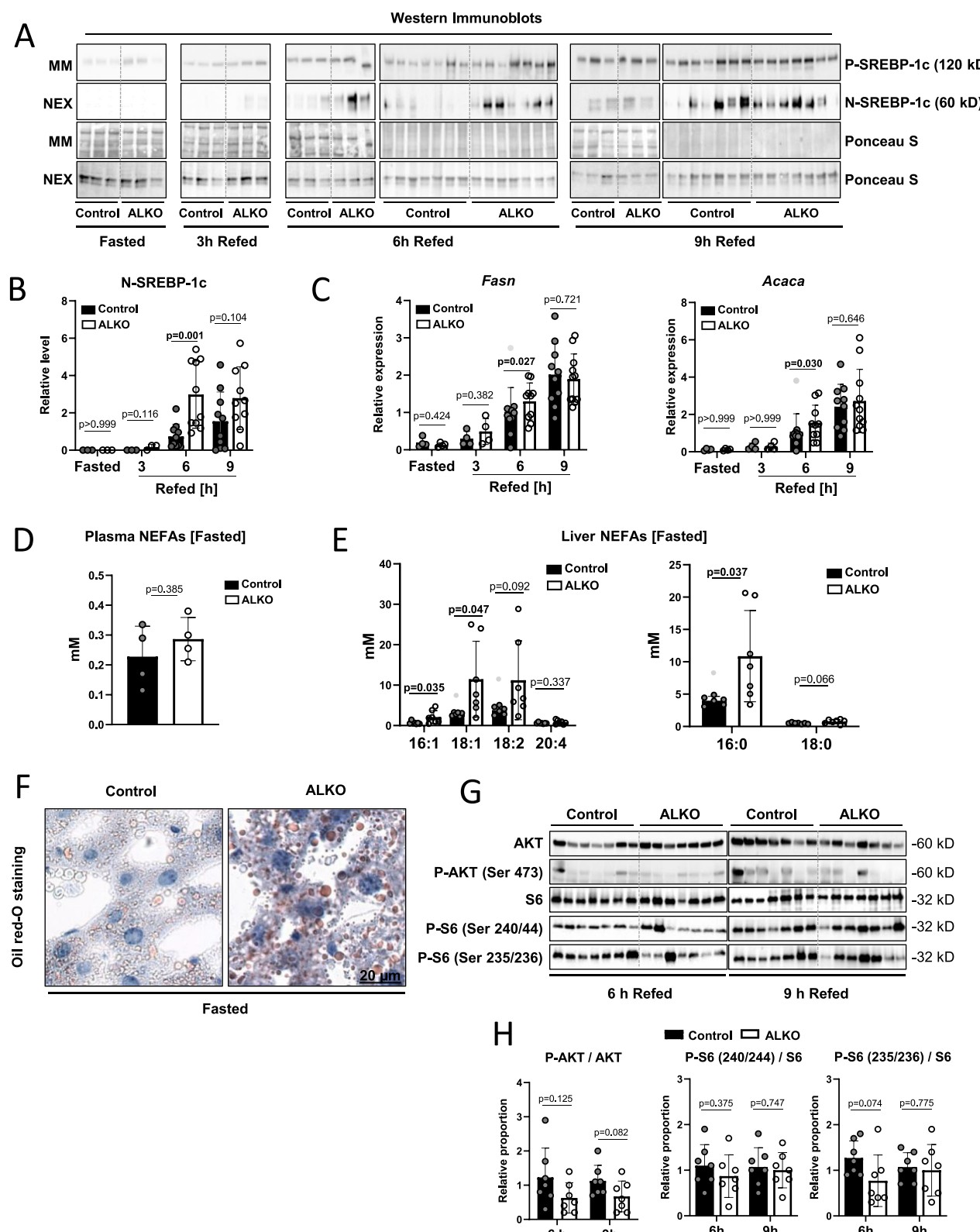

of S6, which was induced by 18:1 injection. However, no difference between the genotypes was observed (Fig. 5E).

Overall, our data indicate that lipolysis derived uFAs suppress P-SREBP-1c cleavage-activation in the liver (Fig. 5B, C). Whether this was a direct effect of the uFAs at the ER membrane, or an indirect effect via modification of plasma insulin-levels could not be determined unequivocally. However, our results on liver specific insulin signaling argue rather against an insulin effect[19]. As a next step, we aimed to directly test if the published molecular model for SREBP cleavage suppression through inhibition of SCAP ER to Golgi transport by uFAs[14,16,26,27] is emulated in mouse hepatocytes.

**Fig. 2 | SREBP-1c processing in mice lacking ATGL in the liver. A** Control and ALKO (liver specific ATGL deficient) mice were fasted for 12 h overnight and subsequently either sacrificed (Fasted) or refed a HChD (high carbohydrate/low-fat diet) and sacrificed at the time points indicated (Refed). Livers were resected and microsomal membrane fractions (MM) and soluble nuclear extracts (NEX) were prepared and subjected to western blot (WB) using a SREBP-1c specific antibody. Ponceau-S stained membranes are shown as loading controls. Each lane represents a liver extract from one mouse. WBs horizontally separated at 6 h and 9 h represent separate experiments. Band intensities were measured using ImageJ, NIH. **B** Levels of proteolytically cleaved N-SREBP-1c were normalized to protein loading and are

presented in the dot plot. **C** Liver *Fasn and Acaca* (mRNA) levels were determined by qPCR (quantitative real-time Polymerase Chain Reaction). **D** Plasma NEFA (non-esterified fatty acids) levels and **E** liver NEFA levels were measured using GC/FID (Gas Chromatography-Flame Ionization Detection), n = 4–7/group. **F** Liver sections were prepared from fasted Control and ALKO mice and stained with Oil-red-O neutral lipid dye. Representative images are depicted. **G** Liver extracts were subjected to WB using the antibodies depicted. **H** Relative proportions of the respective phospho-proteins/proteins were calculated and are presented in the dot plot. n = 7/group. Outliers are shown as light grey dots. Unpaired t-tests were used to compute significance levels.

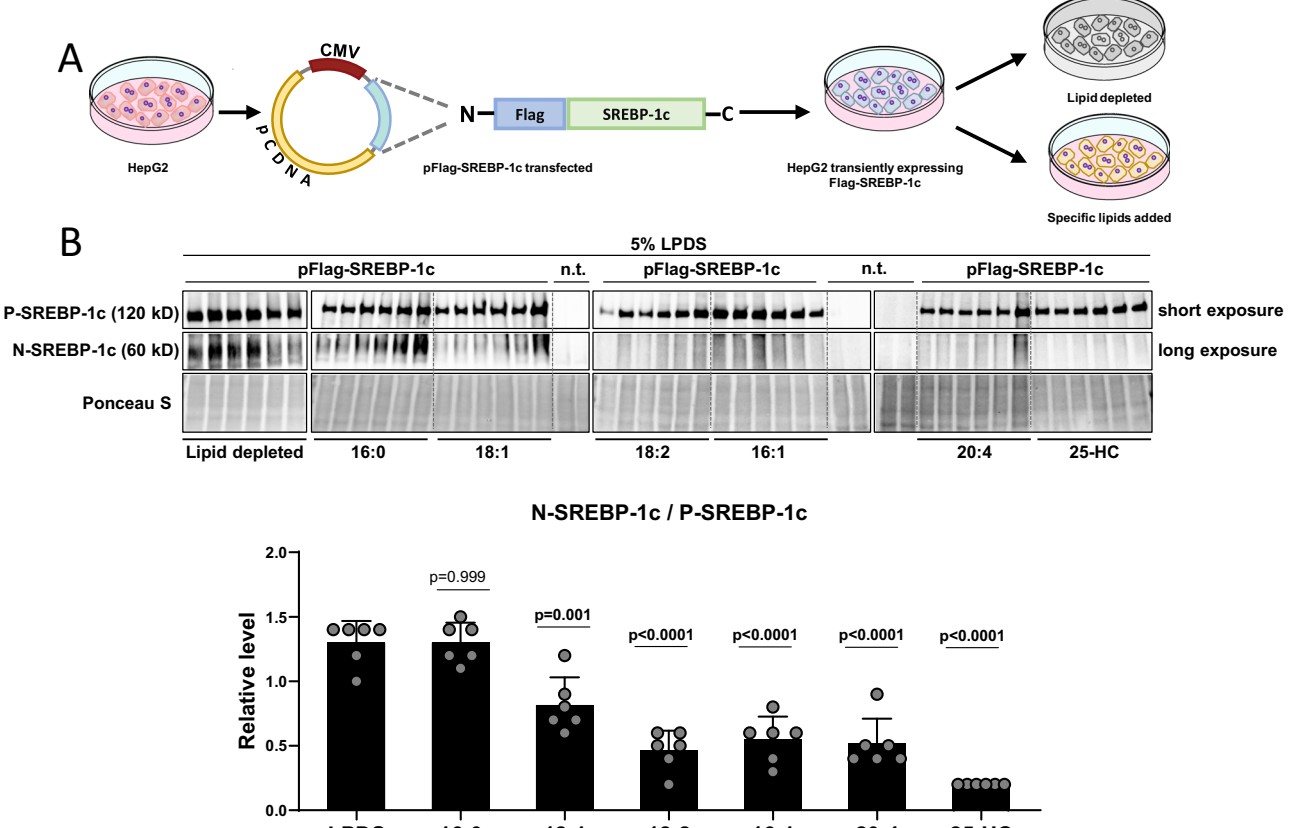

**Fig. 3 | Flag-SREBP1c cleavage reporter vector construction and validation *in vitro*. A** The pCDNA3.1-Flag-SREBP-1c vector contains a triple Flag-tag in front of the human *SREBF1* cDNA (Flag-SREBP-1c) and was transiently transfected into HepG2 cells (Fig. 3A). 48 hours after transfection, HepG2 cells were incubated in lipid depleted medium (5% LPDS) or 5% LPDS plus the addition of 100 μM saturated fatty acids (16:0) or unsaturated fatty acids (18:1, 18:2, 16:1 or 20:4, respectively), or the SREBP-1c suppressor 25 hydroxycholesterol (25-HC), and incubated

for 16 h. Non-transfected HepG2 cells were used as Flag-negative control (n.t). 2 h before harvest, the protease inhibitor ALLN (N-acetyl-leucinyl-leucinyl-norleucinal) was added. **B** Whole cell extracts were subjected to Western Blot (WB). P- and N-SREBP-1c were detected using anti-Flag antibody. Band intensities were measured using ImageJ, NIH. Relative levels of proteolytically cleaved N-SREBP-1c were calculated as the relative fraction of N-SREBP-1c/P-SREBP-1c signal intensities and are presented in the dot plot, n = 6/group.

## Albumin-bound uFAs suppress ER to Golgi transport of SCAP in hepatocytes

uFAs inhibit SREBP-1c cleavage-activation by stabilization of INSIG-1. INSIG-1 anchors the SCAP-SREBP complex in the ER membrane and prevents its migration to the Golgi, where N-SREBP is proteolytically released from P-SREBP. N-SREBP then migrates to the nucleus, where it activates target genes[4,30]. ATGL driven adipose tissue lipolysis releases albumin-bound FAs into the bloodstream. To understand if they regulate SREBP-1c cleavage-activation in liver cells via SCAP- mediated ER to Golgi transport of SREBP-1c, we performed experiments using a previously published pGFP-SCAP vector[30]. This construct allowed us to trace sub-cellular SCAP transport by fluorescence microscopy. Primary hepatocytes were isolated from wild type mice and transfected with pGFP-SCAP.

Subsequently, cells were treated with HPCD [(2-hydroxypropyl)-beta-cyclodextrin] to further deplete membrane lipids[30]. Next, the hepatocytes were either incubated in 5% LPDS medium containing albumin bound-uFA (18:1) or sFA (16:0). Treatment with 25-HC served as SCAP ER to Golgi transport negative control. As readout, we performed co-immunofluorescence microscopy (co-IF) using anti-GFP and anti-GM130 (Golgi-marker) antibodies (Fig. 6A). In accordance with the well-accepted SREBP cleavage-activation model, the addition of 25-HC reduced GFP-SCAP/GM130 co-localization as compared to the lipid depleted condition[26,30]. Similarly, treatment with uFAs showed a significant decrease in GFP-SCAP/GM130 co-localization compared to lipid depletion (Fig. 6B), while sFAs showed no appreciable effect.

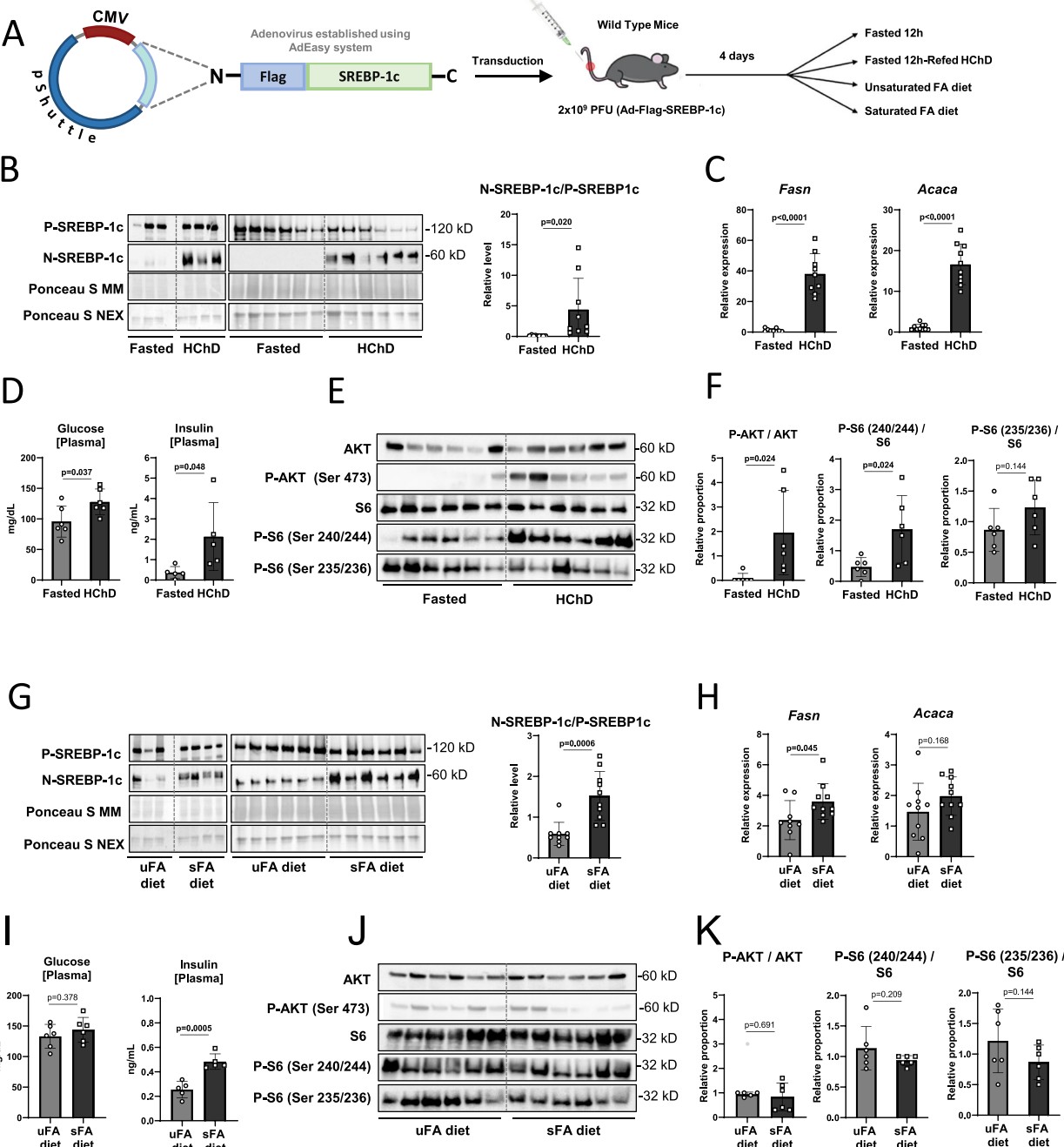

**Fig. 4 | Flag-SREBP1c cleavage reporter vector validation *in vivo*. A** An adenoviral Ad-Flag-SREBP-1c cleavage reporter vector was established using the AdEasy system. Wild type mice were intravenously injected with $2 \times 10^9$ PFU Ad-Flag-SREBP-1c. Four days later, the mice were either fasted overnight (Fasted) or fasted and refed a high carbohydrate diet (HChD) overnight. **B** Thereafter, mice were sacrificed, livers resected, microsomal membrane fractions (MM) and soluble nuclear extracts (NEX) prepared and subjected to WB using an anti-Flag antibody. Ponceau-S stained membranes are shown as loading controls. Each lane represents the liver extract from one mouse. Band intensities were measured using ImageJ, NIH. Relative levels of proteolytically cleaved N-SREBP-1c were calculated as the relative fraction of N-SREBP-1c/P-SREBP-1c signal intensities, normalized to protein loading, and presented in the respective diagrams. **C** Liver *Fasn* and *Acaca* (mRNA) levels were determined by qPCR (quantitative real-time Polymerase Chain Reaction) n = 9–10/group. **D** Blood was drawn and plasma was obtained. Glucose and insulin plasma concentration of fasted and HChD fed mice were determined. Glucose was measured from frozen plasma samples using a glucometer. Plasma insulin levels were determined using a mouse insulin ELISA kit. n = 5–6/group. **E** Liver extracts were subjected to WB using the antibodies depicted. **F** Relative proportions of the respective phospho-proteins/proteins were calculated and are presented in the dot plot. n = 6/group. **G** The experiment was repeated, except that four days after Ad-Flag-SREBP-1c transfection, the mice were fed a diet enriched with flaxseed oil (unsaturated fatty acid diet, uFA) or palm oil (saturated fatty acid diet, sFA) for three consecutive days (see experimental scheme 4**A**). **G** Relative levels of proteolytically cleaved N-SREBP-1c were calculated and are presented in the respective diagrams. **H** Liver *Fasn* and *Acaca* (mRNA) levels were determined by qPCR. n = 9–10/group. **I** Glucose and insulin plasma concentration of uFA- or sFA-rich diet fed mice were determined. n = 5–6/group. **J** Liver extracts were subjected to WB using the antibodies depicted. **K** Relative proportions of the respective phospho-proteins/proteins were calculated and are presented in the dot plot. n = 6/group. Outliers are shown as light grey dots. Unpaired t-tests were used to compute significance levels.

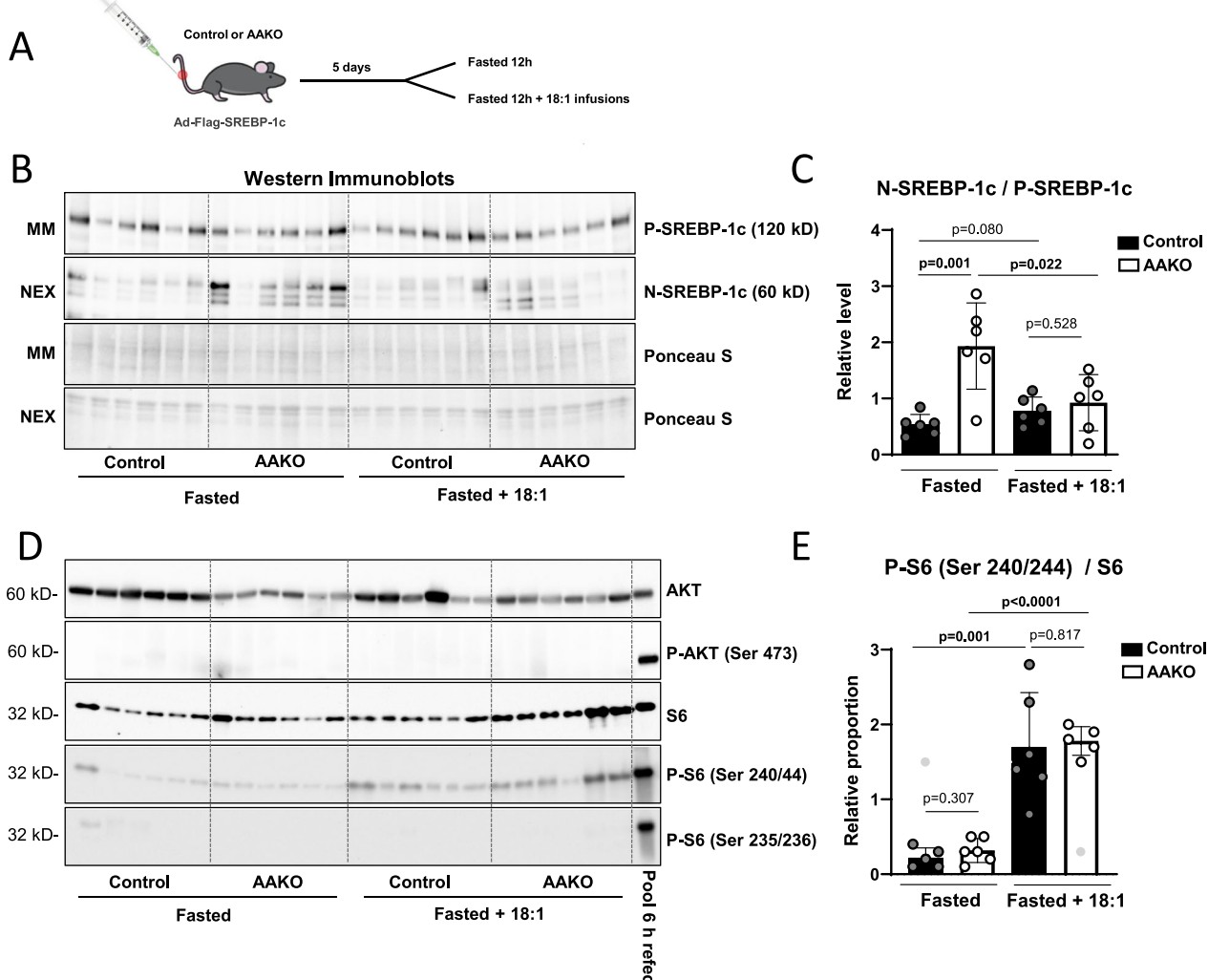

**Fig. 5 | SREBP-1c cleavage-activation is suppressed by lipolysis derived uFAs.**
**A** Control and AAKO (adipose specific ATGL deficient) mice were injected intravenously with $2 \times 10^9$ PFU Ad-Flag-SREBP-1c. Four days later, they were either fasted overnight or fasted overnight and injected with bovine serum albumin complexed oleic acid (18:1) 3 h and 9 h after food withdrawal. All mice were sacrificed in the morning. **B** Livers were resected and microsomal membrane fractions (MM) and soluble nuclear extracts (NEX) prepared and subjected to Western blot (WB) using anti-Flag antibody. Ponceau-S-stained membranes are shown as loading controls. Each lane represents liver

extracts from one mouse. **C** Band intensities were measured using ImageJ, NIH. Relative levels of proteolytically cleaved N-SREBP-1c were calculated as the relative fraction of N-SREBP-1c/P-SREBP-1c signal intensities, normalized to protein loading, and are presented in the dot plot diagram, n = 6/group. **D** Liver extracts were subjected to WB using the antibodies depicted. A pool of 3 extracts from 6 h HChD refed control mice was used as positive control (Pool 6 h refed). **E** Relative proportions of the respective phospho-proteins/proteins were calculated and presented in the dot plot. n = 6/group. Unpaired t-tests were used to compute significance levels.

---

These results indicated that uFAs as well as 25-HC, are able to suppress ER to Golgi transport of SCAP when added to primary mouse hepatocytes that are cultivated under lipid depleted conditions.

## Discussion

We propose that our findings explain one important aspect of the interplay between lipolysis and lipogenesis: ATGL activity in the adipose tissue of fasted animals liberates uFAs that are able to suppress P-SREBP-1c cleavage-activation in the liver for a limited time period after refeeding. As a result, in wild type animals, the FA synthase machinery in the liver is suppressed. Our data in AAKO mice, which lack ATGL specifically in adipose tissue, highlight the transitory nature of this effect (Fig. 1A–E and Supplementary Fig. 2A & B). Accordingly, the SREBP-1c target genes *Fasn*, *Acaca* and *Srebf1* reacted earlier and stronger after refeeding in AAKO compared to controls (Fig. 1C and Supplementary Fig. 1A). However, we wondered why SREBP-1c target genes reacted before N-SREBP-1c was detected by WB. One possible explanation for this is that small amounts of N-SREBP-1c, which are below WB detection limit, may already drive

expression of its target genes. Moreover, Horton et al. were able to detect N-SREBP-1c already 3 h after refeeding using a relatively similar experimental setup[2]. Another question that remained was why SREBP-1c cleavage activation was suppressed in controls as compared to AAKO 6 h after refeeding (Fig. 1A), even though liver uFAs levels had already returned to similar levels as in controls (Fig. 1E & Supplementary Fig. 2B). This may be explained by the relatively slow kinetics of SREBP-1c cleavage-activation in response to exogenous fatty acid levels, which was reported by Hannah et al., in 2001[1]. Finally, it was not clear if adipose derived uFAs directly suppress SREBP-1c activation in the liver, or via modulation of plasma insulin levels (Supplementary Fig. 4B) and modulation of the insulin signaling cascade in the liver (Fig. 1G, H)[19,29]. Therefore, we again tested if ATGL derived FAs regulate SREBP-1c, using ALKO mice, lacking ATGL only in the liver, which showed neither plasma insulin levels (Supplementary Fig. 8B), nor liver insulin signaling (Fig. 2G, H) deregulation. Our results showed that cellular lack of ATGL also leads to SREBP-1c cleavage-activation (Fig. 2A, B). Even though ALKOs have reduced lipolysis capacity in the livers, they showed elevated liver NEFAs (Fig. 2E and Supplementary Fig. 6B). This may sound

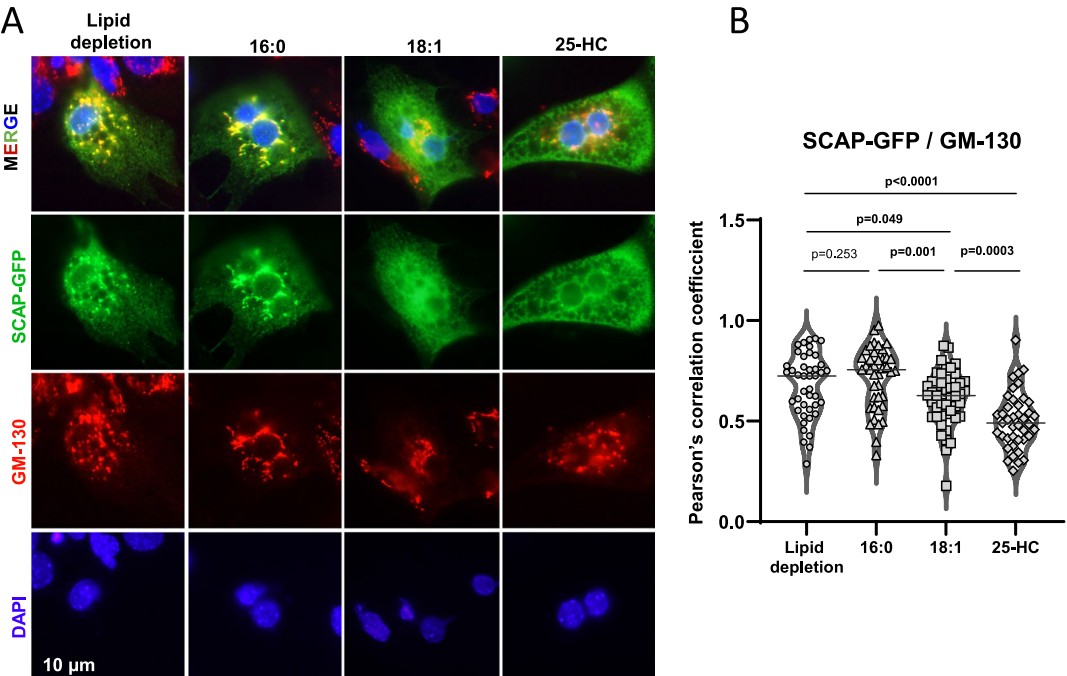

**Fig. 6 | SREBP-1c cleavage-activation is suppressed by lipolysis derived uFAs.**
**A** Primary hepatocytes from wild type mice were transfected with pGFP-SCAP one day after isolation. After 48 h, cells were treated with 1% w/v HPCD [(2-hydroxypropyl)-beta-cyclodextrin] for 1 h. Thereafter, all cells were incubated in 5% LPDS medium with mevalonate and mevastatin present (lipid depletion), with the addition of 100 μM 16:0 or 18:1 FA or with the addition of 2.5 μM of 25-hydroxycholesterol

(25-HC), as indicated. 2 h later, cells were fixed, permeabilized, and GFP-SCAP was visualized by anti-GFP immunofluorescence (IF) (green); Golgi was imaged by anti-GM130 IF (red); DAPI was used for nuclear staining. **B** SCAP-GFP/GM-130 colocalization was determined by Pearson's correlation coefficient, see dot-plots. Unpaired t-tests were used to compute significance levels, n = 40–46 cells/group.

counterintuitive. However, loss of hepatic ATGL leads to reduced PPAR-α signaling mediated FA β-oxidation[18,23], which can explain elevated NEFAs. Moreover, non-parenchymal liver cells with unmodified ATGL expression could also account for the elevated NEFA levels found in the ALKO liver extracts. However, this could not explain the enhanced SREBP-1c activation. One possible explanation is the liver steatosis of ALKO mice (Fig. 2F, Supplementary Fig. 7A, B)[18]. Ferré and Foufelle pointed out a vicious cycle of hepatic steatosis-induced ER stress that activates P-SREBP-1c mediated lipogenesis, which is independent of insulin signaling[25]. Later, the group of Michael Karin showed that ER-Stress in steatotic livers activates SREBP-cleavage via Caspase-2 and S1P, in a SCAP independent manner[24]. However, clarifying this mechanism in ALKO animals was beyond the scope of our manuscript.

Instead, we set out to further test our initial hypothesis that adipose ATGL derived uFAs inhibit P-SREBP-1c cleavage-activation. Earlier publications demonstrated that dietary polyunsaturated FAs inhibit SREBP-1c cleavage-activation in the liver using a viral SREBP-1c cleavage-activation reporter system[28]. To experimentally assess if uFAs, released into the bloodstream by ATGL mediated lipolysis in the adipose tissue, possess the propensity to regulate hepatic SREBP-1c, we constructed and tested an adenoviral Ad-Flag-SREBP-1c cleavage reporter. Ad-Flag-SREBP-1c was modeled after an HSV-tagged construct used to evaluate SREBP-1c regulation in cultured cells by Hua et al.[26]. Our cleavage-activation reporter proved to be suppressed by uFAs in the hepatic cell line HepG2 and in U2OS cells (Fig. 3B & Supplementary Fig. 10)[14–16]. As expected, it was also activated by carbohydrate rich diet feeding after prolonged fasting (Fig. 4B) and suppressed by dietary uFAs in mouse livers (Fig. 4G)[2,28]. During fasting, plasma and liver NEFA concentrations are increased in control mice, due to active ATGL mediated TG hydrolysis in the adipose tissue[7,12]. However, unsaturated NEFA species were diminished in AAKO mice (Fig. 1D, E). With the Ad-Flag-SREBP-1c cleavage-activation reporter in hand, we could directly prove that the absence of adipose tissue ATGL led to enhanced

SREBP-1c cleavage-activation in the liver during fasting. Rescue of this effect by direct injection of albumin-bound uFAs into the bloodstream during fasting, further supports our hypothesis that the uFAs released from adipose tissue during fasting suppress SREBP-1c cleavage-activation in the liver (Fig. 5A–C). However, elevated tissue specific insulin signaling also activates SREBP-1c cleavage[19,29]. Therefore, an alternative explanation for the enhanced SREBP-1c cleavage in fasted AAKOs compared to controls is their elevated glucose tolerance and insulin sensitivity[11,13,17,20]. Reduced plasma glucose levels in fasted AAKOs compared to fasted controls and a relative normalization of this phenomenon upon 18:1 injection may argue in that direction (Supplementary Fig. 13B). However, in the livers, tissue specific insulin signaling was nearly undetectable in all fasted mice and slightly elevated to a similar extent in both AAKOs and controls upon 18:1 injection (Fig. 5D, E). Therefore, we propose that adipose derived uFAs rather suppress SREBP-1c cleavage activation directly at the ER membrane of hepatocytes than indirectly via modification of insulin signaling.

Published in vitro studies demonstrate that uFAs suppress SREBP-1c cleavage-activation in the Golgi-apparatus through anchoring the SREBP chaperone SCAP in the ER. SCAP transport to the Golgi is controlled by the stability of INSIG-1 through a mechanism that requires the ER-associated degradation machinery protein UBXD8[14,15,31]. Our studies of GFP-SCAP trafficking indicate that lipolysis-derived uFAs act through control of SCAP-SREBP ER-to-Golgi transport in primary hepatocytes (Fig. 6A, B). What exactly happens to these free FA on a cellular level, and in which form they act on the SREBP regulatory machinery in the ER is still an open question. One possible explanation could be the incorporation of different FA species into the phospholipids of the ER membrane. The remodeling and FA composition of ER phospholipids could exert differential effects on SREBP-1c processing and activation, as described by Rong et al.[32]. Importantly, the tissue source of these SREBP cleavage-suppressing uFAs was previously unknown. Our studies fill this gap and demonstrate in vivo the importance of adipose derived uFAs and ATGL for homeostatic control of hepatic

lipogenesis. This, however, does not rule out an additional effect caused by changes in insulin sensitivity. In fact, redundancy in the regulation of a biological important mechanism seems possible.

## Materials and methods

### Chemicals

We obtained mevastatin (M2537), mevalonate (50838) and N-acetyl-leucinyl-leucinyl-norleucinal (ALLN, 208719) from Merck, Germany; 25-hydroxycholesterol (25-HC, H1015), lipoprotein-deficient serum (LPDS; S5394), ITS Liquid Media Supplement (100×) (ITS, I3146), Forskolin (F3917) and all other powdered chemical substances from Sigma Aldrich.

### Cell culture media and supplements

Fetal bovine serum (FBS, 10500), high glucose Dulbecco's modified Eagle's medium (DMEM, 41966052) and penicillin-streptomycin (PenStrep, 15140-122) were from Gibco, USA. Standard cell culture medium (D10F) contained, DMEM containing 1x PenStrep and supplemented with 10% (v/v) FBS. Hepatocyte medium: DMEM containing 1x PenStrep, 20% (v/v) FBS, 100 nM dexamethasone, 1x ITS-Supplement. 5% LPDS: DMEM containing 1x PenStrep and supplemented with 5% (v/v) LPDS (Sigma Aldrich, USA, S5394).

### BSA bound FAs

4 mM FA sodium salt (sodium palmitate (Cayman, P9767); palmitoleic acid sodium salt (Sigma Aldrich, USA, 6610-24-8); sodium oleate (Sigma Aldrich, USA, O75011); linoleic acid sodium salt (Sigma Aldrich, USA, L8134) solutions in double distilled $H_2O$, were mixed with 172 mg/ml (w/v) FA free BSA (Sigma Aldrich, USA, A7030) in 2x PBS at 37° under constant vortexing, to achieve BSA bound FA in PBS.

FA infusion into mice was done using BSA-Oleate Monounsaturated 18:1 FA Complex (5 mM) (Item No. 29557, Cayman Chemical).

### Stable Flag-SREBP-1c U2OS cell line generation

We used the human SREBF1 cDNA containing vector pQCXIN (Addgene, USA, 631514) as a template to generate Flag-tagged, full-length, human SREBF1 by conventional PCR, using Flag-hSBP-1c-FW and hSBP-1c-REV primers, listed below. The PCR product was introduced into pCDNA3.1 (Invitrogen, USA, V790-20) using the NEB, USA, builder® HiFi DNA Assembly Cloning Kit (NEB, USA, E5520S). The resulting pCDNA3.1 Flag-SREBP-1c was stably transfected into U2OS cells (ATCC®, USA, HTB96™) using Lipofectamine 2000 (Thermo Fisher Scientific, USA, 11668019) and selected for resistance to G 418 disulfate salt (Sigma Aldrich, USA, A1720). Finally, we generated the UF1c cell line by clonal selection for Flag-SREBP-1c expression. Flag-hSBP-1c-FW: AAGCTTGGTACCGAGCTCGCAC-CATGGATTATAAAGATCATGATATCGATTACAAGGATGACGAT GACAA; hSBP-1c REV: CGGCCGCCACTGTGCTGGATCTAGCTG GAAGTGACAGTGG.

### UF1c cell line maintenance and experiments

UF1c cells were cultured under standard cell culture conditions in a humidified chamber at 37 °C, 5% $CO_2$ in D10F medium. Cells were tested for mycoplasma by qPCR on a regular basis. For experiments, cells were seeded at 60% confluency in 6-well plates. On the next day, cells were washed twice with PBS and subsequently incubated in 5% LPDS for 16 h, containing 100 µM BSA bound FA or 2.5 µM 25-HC. 2 h before harvest, 25 µg/ml ALLN was added. Cells were harvested directly in FSB (final sample buffer: 60 mM Tris-HCl at pH 7,4; 2% (w/v) SDS; 10% glycerol) and subsequently, analyzed by WB.

### HepG2 cell line experiments

HepG2 cells were cultured under standard cell culture conditions in a humidified chamber at 37 °C, 5% $CO_2$ in D10F medium. For experiments, cells were seeded at 70% confluency in 6-well plates. After one day, HepG2 cells were transfected, using Lipofectamine 3000 (Thermo Fisher, L30000-08, Germany) with 1.6 µg of pFlag-SREBP-1c vector containing human SREBF1

cDNA. 48 hours later, cells were washed with PBS and incubated in 5% LPDS, or 5% LPDS plus the addition of 100 µM BSA bound FA or, 2.5 µM 25-HC for 16 h. Two hours before harvest, 25 µg/ml ALLN was added. Cells were harvested directly in FSB (final sample buffer: 60 mM Tris-HCl at pH 7,4; 2% (w/v) SDS; 10% glycerol) and subsequently analyzed by WB.

### Construction of an in vivo SREBP-1c cleavage-reporter vector

The SREBP-1c cDNA was isolated by enzymatic restriction from the pQCXIN vector (Addgene, USA, 631514). Using the AdEasy Adenoviral Vector System (Agilent Catalog #240009), the Flag-SREBP-1c cassette was cloned into a pShuttle-CMV vector. The shuttle vector was co-transformed into BJ5183 cells together with pAdEasy-1. Transformants were selected for kanamycin resistance and recombinants were produced in bulk using the XL10-Gold cell strain. Recombinant plasmid DNA was digested with PacI to expose the Inverted Terminal Repeat and transfected to HEK293 cells for posterior amplification and purification.

### Ethical approval

We have complied with all relevant ethical regulations for animal use. All animal studies were performed in accordance with the guidelines and provisions of the Commission for Animal Experiments of the Austrian Ministry of Education, Science and Research (BMBWF). Approved animal applications and amendments include, BMBWF-66.007/0015-V/3b/2018; BMBWF-66.007/0004-V/3b/2019 and BMBWF-2020-0380.481.

### Animal experiments

Mice were routinely fed ad libitum with a standard chow diet [(4.5% fat, 34% starch, 5.0% sugar and 22.0% protein) M-Z extrudate, V1126, Ssniff Spezialdiäten, Germany]. Mice of "fasted" groups were fasted for 12 h, from 7 p.m. to 7 a.m. Mice in "refed" groups were fasted for 12 h from 7 p.m. to 7 a.m. and then refed a high carbohydrate/low-fat diet (HChD, equivalent to TD 88122; Harlan Teklad, USA) up to 9 h. Mice fed FA-enriched diet were fed ad libitum for 3 consecutive days with an unsaturated FA diet (100 g of chow powder food, 50 g of casein and 60 mL of flaxseed oil) or with a saturated FA diet (100 g of chow powder food, 50 g of casein and 60 mL of palm oil) and subsequently sacrificed. Mice injected with FA were fasted for 12 h from 7 p.m. to 7 a.m. or fasted and intravenously injected with bovine serum albumin complexed oleic acid (18:1, BSA-Oleate Monounsaturated FA Complex (5 mM) Item No. 29557, Cayman Chemical) 3 and 9 h after food withdrawal. Mice were sacrificed 3 h after the second FA infusion.

Mouse strains used: WT: C57Bl/6 J (own breeding, originally from Jackson lab). Genetically modified strains on C57Bl/6 J background: AAKO, Adipose-tissue specific Atgl-knockout (Atgl^flox/flox, Adipoq-Cre)[17]; ALKO, Liver specific Atgl-knockout (Atgl^flox/flox, Alb-Cre)[18].

### Hepatocyte isolation

Primary hepatocytes were isolated by perfusion of mouse livers with 40 ml perfusion buffer (5.5 mM KCl, 0.1% Glucose, 2.1 g/l NaHCO₃, 700 µM EDTA, 10 mM Hepes and 150 mM NaCl). After 20 min the buffer was exchanged to 50 ml collagenase buffer (5.5 mM KCl, 0.1% Glucose, 2.1 g/l NaHCO₃, 10 mM Hepes and 150 mM NaCl, 3.5 mM CaCl₂, 1% BSA, 500 µg/ml Collagenase Type I (300 U/mg)). Livers were perfused at a perfusion rate of 2 ml/min. Subsequently, livers were dissociated with a plunger of a syringe in 10 ml D10F, applied onto a 100 µM cell strainer, and the flow through collected. Hepatocytes were centrifuged at $100 \times g$ for 2 min and washed twice in DMEM containing 1x PenStrep. Next, cells were re-suspended in hepatocyte medium. Cells were counted after staining with trypan blue (Thermo Fischer Scientific, USA, 15250061) to assess viability. Primary hepatocytes were seeded in rat tail collagen I (Sigma Aldrich, USA, C3867) coated 6-well dishes at a density of $2.5*10^5$ cells/well.

### Immunofluorescence (IF)

IF was performed, and co-localization of GFP-SCAP and GM130 was analyzed, as previously described by Shao et al., with modifications as

follows[30]. Hepatocytes were seeded into 1 well of a 6 well-plate containing a sterile coverslip using standard hepatocyte medium at a density of $2.5*10^5$ cells/well. Hepatocytes were transfected one day after isolation using Lipofectamine 2000 (Thermo Fisher, USA, 11668019) with 1 µg pGFP-SCAP plasmid. 48 h post transfection cells were treated with 1% hydroxypropyl-beta-cyclodextrin (HPCD) to deplete sterols for 1 h in plain DMEM medium. Next, cells were refed for 2 h with 5% LPDS medium plus the addition of 100 µM 16:0 or 18:1 BSA-bound FAs or 2.5 µM of 25-hydroxycholesterol (25-HC), with 50 µM mevalonate and 50 µM mevastatin. Thereafter, cells were washed with PBS twice, fixed in formaldehyde/ PBS (0.03/1; w/v) at room temperature for 10 min and then permeabilized with Triton X-100/PBS/glycine (0.05/0.9/0.1; v/v/v) for 3 min at room temperature. Next, cells were incubated for 30 min with primary antibodies (anti-GFP and anti-GM130) and respective (Alexa-488 (green) and Alexa-594 (red) coupled) secondary antibodies, followed by DAPI (Sigma Aldrich, USA, D9542) staining. For details, see Table 1. Coverslips were mounted onto glass-slides and dried in the dark overnight before visualization by an Olympus BX51 microscope at the microscopy core facility of the ZMF, MUG, Graz, Austria. Quantitative co-localization analysis was conducted using 40–45 cells per condition. The analysis was performed using Image J with the JACoP plug-in[30].

## Liver microsomal and nuclear fractionations

Liver cell fractions were prepared as previously described, with minor modifications[33]. Livers were excised and washed in ice-cold PBS and immediately frozen in liquid nitrogen cooled methylbutane. To isolate nuclear fractions, 600 mg of frozen liver was mixed with 6 ml buffer A (10 mM Hepes at pH 7.6, 25 mM KCl, 1 mM sodium EDTA, 2 M sucrose, 10% vol/vol glycerol, 0.15 mM spermine, 2 mM spermidine, 1x protease Inhibitor cocktail, 50 µg/ml ALLN). Livers were homogenized by three strokes with a Teflon pestle in a potter homogenizer at low speed. The homogenate was filtered through a 100 µM cell strainer. Samples were overlaid with 2 ml buffer A in SW 41Ti tubes, which were filled with Buffer 1 (10 mM Hepes at pH 7.6, 25 mM KCl, 1 mM sodium EDTA). Then, the samples were centrifuged at 25,000 rpm (75,000 × g) for 1 h at 4 °C in a SW 41 Ti Rotor. Subsequently, the tubes were turned over, the supernatant was collected for later use to isolate microsomal membrane extracts, and the nuclear pellet was recovered from the bottom and resuspended in 1 ml buffer D (10 mM Hepes pH 7.6, 100 mM KCl, 2 mM $MgCl_2$, 1 mM sodium EDTA, 10% (vol/vol) glycerol, 1 mM DTT, 1x protease inhibitor cocktail, 50 µg/ml ALLN). For extraction of soluble nuclear proteins, 140 µl buffer AS (3.3 M ammonium sulfate (pH 7.9) was added to the solution, agitated gently for 40 min at 4 °C on a rotating wheel in a cold room and subsequently centrifuged at 78,000 rpm in Beckman TLA-100.4 rotor for 45 min at 4 °C. The supernatant was mixed 1:5 with 5x FSB and designated soluble nuclear protein extract (NEX). For membrane fractions isolation, 1 mL from the supernatant previously collected after the first centrifugation cycle was resuspended in 9 mL buffer M (20 mM Tris-HCl pH 7.4, 2 mM $MgCl_2$, 0.25 mM sucrose, 10 mM sodium EDTA, 10 mM sodium EGTA, 1x protease inhibitor cocktail, 50 µg/ml ALLN). The homogenate was centrifuged at 30,000 rpm for 1 h at 4 °C in a SW 41 Ti Rotor. The resulting pellet was dissolved in 1x FSB and designated cytosolic microsomal membrane extract (MM).

## Blood biochemistry

Plasma NEFA levels were analyzed using the NEFA kit HR Series NEFA-HR (2) (276-76491, 995-34791, 993-35191, 999-34691, 991-34891, WAKO Chemicals, Japan) according to manufactures instructions. Liver TG levels were measured from liver Folch extracts[34] using the Triglycerides FS 10 kit (Diasys, Germany, 15760991002). Samples were dissolved in 600 µl 1% Triton X-100.

Plasma insulin concentration was determined using the mouse Enzyme-linked Immunosorbent Assay (ELISA) test (Crystal Chem, USA # 90080). Plasma glucose concentration was measured from frozen plasma samples using a glucometer. In some cases, where only blood was available

for glucose measurement we used a conversion method previously described[35].

## Fatty acid analysis

Livers were homogenized, subjected to Folch extraction and lipids dried under a stream of nitrogen. Lipid extracts were pre-separated by thin layer chromatography (TLC). The band co-migrating with a triolein standard was scraped off and after addition of C15:0 as internal standard directly trans-esterified (1.2 ml toluene and 1 ml boron trifluoride-methanol (20%)) at 110 °C for 1 h. Gas Chromatography analysis of the corresponding FA methyl esters was performed as described[12] and concentrations were quantitated by peak area comparison with the internal standard.

## Western blot (WB)

WB was performed using 4–20% SDS gels (Bio-Rad, 4561096) and blotted on 0.45 µm nitrocellulose membranes (GE Healthcare, 15259794). Next, membranes were stained with Ponceau-S solution (Sigma Aldrich, USA, P7170) for protein load quantification and subsequently blocked using skim milk powder (Sigma Aldrich, USA, 1153630500)/PBS/TWEEN® 20 (Sigma Aldrich, USA, P1379) (0.05,1,0.05; w/v/v). Membranes were incubated with the antibodies mentioned in the respective figures, which are described in detail in Table 1. As secondary antibodies, respective horseradish peroxidase coupled secondary antibodies were used (see Table 1). Signals were detected using the ChemiDoc Imaging System (Biorad, USA, 17001401). WB band intensities were analyzed using image J, NIH, software package[36].

## Quantitative real-time polymerase chain reaction (qPCR)

RNA was isolated from homogenized livers using Trizol (Invitrogen, USA, 15239794), cDNA prepared using High-Capacity cDNA Reverse Transcription Kit (Applied Biosystems, USA, 4368814) and qPCR performed using the SYBR Green Luna® Universal qPCR Master Mix (NEB, USA, M3003) on the QuantStudio™ 7 Flex Real-Time PCR System (Applied Biosystems™, USA, 4485701). Primers were designed with the NCBI primer designing tool (primer-blast) and are listed in Table 2. Relative mRNA levels were analyzed as described by Schmittgen et al.[37]. The relative PCR-efficiency of each primer set used was determined by computational standard curve analysis in QuantStudio™ 7 Flex Software, Applied Biosystems™, USA. Only primer sets showing efficiency between 90% and 105% were used. Relative gene expression was computed using the delta delta CT method using primers against 18 s rRNA as internal standard.

## Oil-Red-O staining

Oil-Red-O (ORO-staining) (Sigma-Aldrich, USA, O0625) was performed on 4% neutral buffered formalin fixed, frozen sections from mouse livers that were fresh frozen in $LN_2$ cooled methylbutane.

## Statistics and reproducibility

Each biological replicate was defined as a biological unit (e.g., a cell, a liver, a mouse). If applicable, biological replicate values were computed as the arithmetic mean value of technical replicate values. All data represent mean values of the biological replicates ± standard error. The number of biological replicates for each experiment is reported in the figure legends. Statistical analysis between 2 groups of biological replicates were performed using Student's 2-tailed t-test, using GraphPad Prism, version 8.2.0. Outlier analysis was performed using Grubb's test in GraphPad (https://www.graphpad.com/quickcalcs/grubbs1/), with α = 0.05. Linear correlations were determined by Pearson's correlation coefficient. (Holm Sidak method, alpha 0.05).

## Graphics

The statistical graphics were created using GraphPad Prism, version 8.2.0. Part of the illustrations used in the figures were sourced under a free license from Freepik (Image by nikapeshkov on Freepik: https://de.freepik.com/vektoren-kostenlos/sammlung-verschiedener-medizinischer-spritzen-fuer-impfstoffe_19286276.htm).

## Table 1 | Resources table

| Resources table | | | | |
|---|---|---|---|---|
| **Reagent type** | **Designation** | **Source or reference** | **Identifiers** | **Additional information** |
| Antibody | Anti-SREBP-1c, clone 2A4. (Mouse monoclonal) | Abcam | ab3259 | (1:500) |
| Antibody | Anti-SREBP-1c clone 20B12. (Rabbit monoclonal) | Merck | MABS1987 | (1:500) |
| Antibody | Anti-FLAG® M2 (Mouse monoclonal) | Sigma-Aldrich | F3165 | (1:500) |
| Antibody | Anti-FLAG® M2-Peroxidase (HRP) (Mouse monoclonal) | Sigma-Aldrich | A8592 | (1:500) |
| Antibody | Anti-GFP antibody. (Rabbit polyclonal) | Abcam | ab290 | (1:500) |
| Antibody | Anti-GM130, clone 35. (Mouse monoclonal) | BD Biosiences | 610822 | (1:250) |
| Antibody | Anti-Rabbit IgG (H + L), Alexa Fluor 488 coupled. (Goat polyclonal) | Invitrogen | A11034 | (1:250) |
| Antibody | Anti-Mouse IgG (H + L), Alexa Fluor 594 coupled. (Mouse polyclonal) | Invitrogen | A11005 | (1:250) |
| Antibody | Anti-mouse Immunoglobulins/HRP. (Goat polyclonal) | Dako | P0477 | (1:3000) |
| Antibody | Anti-rabbit Immunoglobulins/HRP (Pig polyclonal) | Dako | P0217 | (1:3000) |
| Antibody | Anti-S6 ribosomal protein, clone 5G10 (Rabbit polyclonal) | Cell Signaling | AB_331355 | 1:1000 |
| Antibody | Anti-phospho-S6 Ribosomal Protein, (Ser240/244) (D68F8) (Rabbit polyclonal) | Cell Signaling | AB_10694233 | 1:1000 |
| Antibody | Anti-phospho-S6 Ribosomal Protein, (Ser235/36) (Rabbit polyclonal) | Cell Signaling | AB_10858004 | 1:1000 |
| Antibody | Anti-AKT (Rabbit polyclonal) | Cell Signaling | AB_329827 | 1:1000 |
| Antibody | Anti-phospho-Akt, (Ser473), clone D9E (Rabbit polyclonal) | Cell Signaling | AB_2315049 | 1:1000 |
| Cell line | U2OS | ATCC | HTB96 | |
| Cell line | HepG2 | ATCC | HB-8065 | |
| Strain/Strain background (*Mus musculus*) | AAKO (*Atgl*flox/flox, *Adipoq-Cre*) | Rudolf Zechner | JAX: 024278 x JAX: 028020 | Adipose specific *Atgl* knockout mice; own breeding |
| Strain/Strain background (*Mus musculus*) | ALKO (*Atgl*flox/flox, *Alb-Cre*) | Rudolf Zechner | JAX: 024278 x JAX: 018961 | Liver specific *Atgl* knockout mice; own breeding |

## Table 2 | qPCR primer sequences

| Primer pair | Forward | Reverse |
|---|---|---|
| *mFasn* | GGCCCCTCTGTTAATTGGCT | GGATCTCAGGGTTGGGGTTG |
| *mAcaca* | GGCCAGTGCTATGCTGAGAT | CCAGGTCGTTTGACATAATGG |
| *m18s* | GTAACCCGTTGAACCCCATT | CCATCCAATCGGTAGTAGCG |
| *mHmgcr* | CAACCTTCTACCTCAGCAAGC | CACAGTGCCACATACAATTCG |
| *mHmgcs* | GTTCCCTGGCTTCTGTCCT | CCAAGCCAGAACCGTAAGAG |
| *mSrebf1c* | GGAGCCATGGATTGCACATT | GGCCCGGGAAGTCACTGT |
| *mEnd_Srebf1c* | CTGACAGGTGAAATCGGCG | AATCCATGGCTCCGTGGTC |

### Reporting summary

Further information on research design is available in the Nature Portfolio Reporting Summary linked to this article.

### Data availability

The source data underlying the graphs in the main and Supplementary Figs. (Supplementary data 1 and Supplementary data 2, respectively) are uploaded to Figshare: https://figshare.com/s/5583fdf547834d503bb8. Uncropped and unedited blot images are displayed in Supplementary Fig. 14. For any questions regarding material or methods, please contact the corresponding author.

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

## Acknowledgement

The authors would like to thank Jared Rutter for kindly providing *SREBP-1c* cDNA; Wei Shao, Chiaki Ishida, Chune Liu, Debaditya Mukhopadhyay and Shan Zhao for their continued assistance and advice during experiments done in Dr. Espenshade's laboratory; Peter Hofer, Guenter Haemmerle and Kathrin Zierler for technical assistance and critical advice; Markus Absenger for technical assistance. We thank Grazyna Kwapiszewska-Marsh and Leigh Marsh for their continued support and critical discussion of our work. This research was funded in whole or in part by the Austrian Science Fund (FWF) Metabolic and Cardiovascular Disease (DK-MCD) Program [grant 10.55776/W1226]. For open access purposes, the author has applied a CC BY public copyright license to any author-accepted manuscript version arising from this submission. Additional funding was provided by an FWF grant (10.55776/PAT9976323) and the Medical University of Graz. Peter J

Espenshade was funded by the National Institutes of Health (grants HL077588 and GM149312). Beatrix Wieser and Paola Pena de la Sancha received additional funding through the Austrian Marshall Plan Scholarship Program (953 1175 38 22 2019). Beatrix Wieser received additional funding from the Bundesministerium fuer Bildung, Wissenschaft und Forschung (BMBWF), Oesterreichische Austausch Dienst-GmbH (OeAD-GmbH), Zentrum fuer Internationale Kooperation und Mobilitaet (ICM) through a Marietta Blau Grant (ICM-2019-13518). Paul Willibald Vesely was supported by a European Research Council Grant, LipoCheX (340896), by an FWF grant (10.55776/P30968) and is currently employed at the Medical University of Graz, Austria via an FWF grant (10.55776/KLI-844B). Dagmar Kratky and Rudolf Zechner were supported by the SFB Lipid Hydrolysis (10.55776/F73).

## Author contributions

P.P.S., B.I.W., P.W.V., G.H., M.S., R.Z., R.B. and P.J.E. contributed to experimental design, data interpretation and manuscript preparation. P.P.S., B.I.W., P.W.V., and S.S. performed and analyzed qPCR data. P.P.S and B.I.W. performed cell culture experiments and WBs. P.P.S., B.I.W., P.W.V., and S.S. conducted in vivo studies. S.S., B.I.W., and P.W.V. performed immunofluorescence. P.S and B.I.W. performed hepatocyte experiments. P.P.S and B.I.W. biochemically analyzed blood samples. S.S. performed ORO-staining. D.K., W.S., and H.R. performed FID/GC. P.P.S., B.I.W., and P.W.V. performed statistical analysis. P.P.S. and P.W.V. generated Ad-Flag-SREBP-1c cleavage-reporter vector. P.P.S, S.F., and M.L. purified and amplified Ad-Flag-SREBP-1c cleavage-reporter vector.

## Competing interests

The authors declare no competing interests.
