## [Transparent Peer Review file · Communications Biology]

Lipolysis-derived fatty acids are needed for homeostatic control of Sterol Element-Binding Protein-1c driven hepatic lipogenesis.

Corresponding Author: Dr Paul Vesely

Version 0:

Reviewer comments:

Reviewer #1

(Remarks to the Author)

Based on previous publication showing that unsaturated fatty acids (uFAs) inhibit the proteolytic cleavage of SREBP-1c, the authors aim to demonstrate that uFAs produced by adipose tissue lipolysis during fasting play an important role in the inhibition of hepatic lipogenesis by blocking SREBP-1c activation. Using mice invalidated for ATGL (adipose triglyceride lipase) in adipocytes (wat ATGL KO Mice), one of the main lipolytic enzyme, they show that plasma and liver fatty acid concentrations and especially uFAs are decreased in ATGL KO mice. This is concomitant with a higher activation of hepatic SREBP-1c at refeeding. Then, the authors repeat the experiments but in mice invalidated for ATGL in hepatocytes (liver ATGL KO Mice). They show that SREBP-1c activation is very heterogeneous between animals and conclude that liver ATGL has no clear effect on SREBP-1c activation.

The authors develop then several models in which they express the precursor SREBP-1c form with a flag tag (cell lines stably expressing pSREBP-1c or adenovirus producing tagged form version of full length SREBP-1c. These tools are particularly interesting to answer to their questions

Major points : the authors miss important findings concerning the regulation of SREBP-1c. It was initially shown by the group of Brown and Goldstein that this isoform which is highly expressed in hepatocytes does not respond as the other isoforms SREBP-1a and SREBP-2 to cholesterol depletion (Shimomura et al Proc Natl Acad Sci U S A. 96:13656-61.1999). Since then, it has been repeatedly demonstrated by various teams that insulin is the main regulator of SREBP-1c activation at refeeding (at both transcriptional and post-translational levels). The role of insulin must be taken into account by the authors (see the points below). The authors cannot ignore these results. It is therefore irrelevant to test both in vitro and in vivo the activation of SREBP-1c in response to cholesterol depletion.

1-Given the crucial role of insulin in SREBP-1c activation, it is important to provide insulin concentrations in the WAT ATGL KO mice and to verify that hepatic insulin signaling is not modified in these animals .

2- The use of U2OS cells is not adequate to assess the activation of SREBP-1c by insulin, an hepatocyte cell line would have been preferable. For that reason, the authors have tested whether cholesterol concentration modulates the cleavage which is not relevant with the physiological regulation of SREBP-1c

3- Since SREBP-1c expression is regulated by a feed forward mechanism, the expression of sreb-1c (mRNA) must be measured in all experiments.

4- the experiments with the adenovirus expressing SREBP-1c full length form in vivo are more convincing since the authors nicely show that their construct is activated at refeeding with a carbohydrate diet demonstrating the key role of insulin in this activation. However, once again, the authors should show that the refeeding with diets enriched in uFAs and saturated fats has not modified the response to insulin or insulin secretion in these animals.

Reviewer #2

(Remarks to the Author)

In this study, Peña de la Sancha and colleagues tested the hypothesis that unsaturated fatty acids released by ATGL-

mediated lipolysis in adipocytes are important regulators of SREBP cleavage and activation. Their main findings indicate that mice deficient in ATGL in adipocytes have a faster recovery of liver SREBP1 cleavage and activation upon carbohydrate refeeding, such effect that was attributed to a reduced lipolysis and serum unsaturated fatty acid content. Authors perform a series of experiments to prove that adipocyte derived unsaturated fatty acids are in fact modulators of liver SREBP cleavage. Please find below some suggestions to help manuscript convey its message.

n of 3 is unacceptable and inappropriate for statistical analysis. Please increase n.

Please report serum insulin and glucose levels for the fasting-refeeding experiment reported in Figure 1 and 2 and different diets reported in Figure 4C. Is it possible that the Atgl ko mice due to low serum fatty acid have a lower insulin level than controls upon fasting and carbohydrate refeeding?

Figure 3E - If adipose tissue is the main source of unsaturated fatty acids for SREBP1 cleavage inhibition and ATGL has a preference for the hydrolysis of unsaturated fatty acids, short-term feeding (3 days) with diets composed of different fatty acids should not affect serum fatty acid composition and SREBP1 cleavage, unless they promote a strong depletion in unsaturated fatty acids in adipose tissue, which is not the case. Please show serum fatty acid profile for this experiment.

Figure 4 – is it possible that the injected oleic acid is restoring fasting insulin levels in Atgl ko mice that inhibits the srebp cleavage?

Please explain why you depleted membrane lipids in the experiments reported in Figure 4D and E. Is this physiological?

Considering the redundant role of ChREBP in the regulation of de novo lipogenesis, does this faster recovery of SREBP1 cleavage on atgl ko mice upon refeeding leads to higher rates of liver de novo lipogenesis than controls? Is this mechanism of regulation relevant considering that de novo lipogenesis is markedly decreased upon fasting where

How do you conciliate your hypothesis with the increase in SREBP and de novo lipogenesis seen upon insulin resistance in which lipolysis is markedly elevated in adipose tissue?

Version 1:

Reviewer comments:

Reviewer #1

(Remarks to the Author)

Thanks to the authors for adding the results on insulin concentrations and its signaling. However, given the heterogeneity of the samples in the fed state (fig1G), it would have been more judicious to present the fasting state

Reviewer #2

(Remarks to the Author)

Authors have addressed some of my previous concerns, but manuscript still contains several inconsistencies that need to be addressed specially regarding a role of insulin regulation srebp cleavage. In fact, either changes in insulin or insulin signaling could, in all experiments, explain the changes in SREBP cleavage.

Abstract

Lines 9-11- There is a transitory increase in liver SREBP after 6h in adipocyte atgl Ko mice, but not 9 h of refeeding. Therefore, the term hyper-activated is exaggerated. Please describe findings specifically.

Line 14-15- It is important to mention that fatty acids seem to play an inhibitory role on SREBP cleavage mainly upon refeeding, not upon fasting.

Lines 16-19 “This homeostatic regulation...” Please delete this. Too much speculation.

Results

Figure 1 - There is a transitory increase in liver N-SREBP content after 6h in adipocyte ATGLKO mice, but not 3 h or 9 h of refeeding. Upon fasting, there are no differences in N-SREBP-1c, FASN and ACACAC between genotypes. After refeeding for 3 h, SREBP is still at levels found upon fasting and there are no differences between genotypes, but Srebf, Fasn and Acaca mRNA levels are higher in adipocyte ATGLKO. This indicates that SREBP is not driving this increase in FASN and ACACA mRNA levels found upon 3 h of refeeding, as it requires cleavage for activation. Furthermore, you are showing a reduction in serum fatty acid levels between genotypes upon fasting, but the difference in SREBP occurs only after 6 h of refeeding. At this time point (6h refeeding), serum fatty acids are higher in adipocyte ATGLKO than controls and there are no changes in serum unsaturated fatty acids, and liver fatty acid or TAG contents between genotypes. These findings go against your hypothesis and conclusions.

Why adipocyte ATGLKO display higher Srebf mRNA levels at fasting and SREBF, FASN and ACACA mRNA upon 3 h refeeding?

In panel G, it is impossible to truly evaluate phospho Akt results. Upon 6 h refeeding, there are 3 adipocyte ATGLKO mice that show levels higher than at least 5 controls, but in the remaining 4 adipocyte ATGLKO mice there is a white blur indicative of issues in protein transference or blotting revelation. Similar issues are found in phospho Akt at 9 h. The only convincing blotting result is the increase in phospho S6 content in liver upon 6 h refeeding, indicating a major role of insulin promoting SREBP cleavage at this time point, not a reduction in unsaturated fatty acids.

Lines 98-106 – results do not support any role for fatty acids, but a clear role of insulin via S6K/S6 promoting SREBP cleavage at 6 h refeeding. This could be further tested by acutely inhibiting mTORC1 with rapamycin.

Figure 2. The findings that mice bearing liver atgl deletion reproduces the same increase in N-SREBP upon 6 h refeeding shown by mice with adipocyte atgl deletion is puzzling specially because this is not associated with an increase in phospho S6 content as it does upon adipocyte atgl deletion.

Again in Figure 2, phospho Akt blots are of very low quality. Why atgl deletion in either hepatocytes or adipocytes results in a similar SREBP phenotype?

In Figure 3, the inhibitory effect on SREBP cleavage of diet enriched with unsaturated fatty acids occurred along a reduction in serum insulin levels, which complicates the interpretation. This insulin result should be in Figure 3, not supplemental.

Again in Figure 13 supplemental, phospho Akt blots are of very low quality, which precludes an evaluation of insulin signaling.

In Figure 4, higher SREBP cleavage occurred in association with higher serum insulin levels in adipocyte atgl ko mice and, 18:1 administration reduced both SREBP cleavage and serum insulin levels. Again, changes in serum insulin could explain all the changes in SREBP.

Version 2:

Reviewer comments:

Reviewer #2

(Remarks to the Author)

Authors have properly addressed my previous comments. I have no further comments.

Dear Reviewers,

Please find our answers to your requests and questions below. For your convenience, we pasted the most relevant figures and legends into this letter.

Reviewers' comments:

Reviewer #1 (Remarks to the Author):

Based on previous publication showing that unsaturated fatty acids (uFAs) inhibit the proteolytic cleavage of SREBP-1c, the authors aim to demonstrate that uFAs produced by adipose tissue lipolysis during fasting play an important role in the inhibition of hepatic lipogenesis by blocking SREBP-1c activation. Using mice invalidated for ATGL (adipose triglyceride lipase) in adipocytes (wat ATGL KO Mice), one of the main lipolytic enzyme, they show that plasma and liver fatty acid concentrations and especially uFAs are decreased in ATGL KO mice. This is concomitant with a higher activation of hepatic SREBP-1c at refeeding. Then, the authors repeat the experiments but in mice invalidated for ATGL in hepatocytes (liver ATGL KO Mice). They show that SREBP-1c activation is very heterogeneous between animals and conclude that liver ATGL has no clear effect on SREBP-1c activation.

The authors develop then several models in which they express the precursor SREBP-1c form with a flag tag (cell lines stably expressing pSREBP-1c or adenovirus producing tagged form version of full length SREBP-1c. These tools are particularly interesting to answer to their questions

Major points: the authors miss important findings concerning the regulation of SREBP-1c. It was initially shown by the group of Brown and Goldstein that this isoform which is highly expressed in hepatocytes does not respond as the other isoforms SREBP-1a and SREBP-2 to cholesterol depletion (Shimomura et al Proc Natl Acad Sci U S A. 96:13656-61.1999). Since then, it has been repeatedly demonstrated by various teams that insulin is the main regulator of SREBP-1c activation at refeeding (at both transcriptional and post-translational levels). The role of insulin must be taken into account by the authors (see the points below). The authors cannot ignore these results. It is therefore irrelevant to test both in vitro and in vivo the activation of SREBP-1c in response to cholesterol depletion.

1-Given the crucial role of insulin in SREBP-1c activation, it is important to provide insulin concentrations in the WAT ATGL KO mice and to verify that hepatic insulin signaling is not modified in these animals.

Dear Reviewer, we agree that SREBP-1c is transcriptionally and post-translationally regulated by insulin signaling. Therefore, we measured plasma glucose and insulin concentrations in samples of all animal experiments and performed liver insulin signaling-

relevant western immunoblots in the most important experimental conditions. See below the requested analyses on WAT ATGL KO mice (AAKO), plasma insulin-levels, supplementary figure 4B, and analysis of hepatic insulin signaling, figure 1G & H. As mentioned in the manuscript text, we conclude that, “The enhanced tissue specific insulin sensitivity in the liver may have an additional positive effect on P-SREPB-1c cleavage activation and activation of FA biosynthesis in AAKO when compared to controls”.

Supplementary Figure 4. AAKO Plasma glucose and insulin. Groups of Control and AAKO (adipose specific ATGL deficient) mice were fasted for 12 h overnight and subsequently, either sacrificed (Fasted) or refed a HChD (high carbohydrate/low fat diet) and sacrificed at the time points indicated (Refed). Blood was withdrawn and plasma was obtained. (A) Plasma glucose concentrations were measured in frozen samples using a glucometer. (B) Plasma insulin levels were determined using a mouse insulin ELISA kit. $n=4-12$ / group. Unpaired T-tests were used to compute significance levels.

G**H**
Figure 1. AAKO Insulin signaling Control and AAKO (adipose specific ATGL deficient) mice were fasted for 12 h overnight and subsequently either sacrificed (Fasted) or refeed a HChD (high carbohydrate/low-fat diet) and sacrificed at the time-points indicated (Refed). Livers were resected, microsomal membrane fractions (MM) were prepared and subjected to Western Blot (WB). **(G)** Liver extracts were subjected to WB using the antibodies depicted. **(H)** Relative proportions of the respective phospho-proteins / proteins were calculated and are presented in the dot plot. n=6 / group. Outliers are shown as light grey dots. Unpaired T-tests were used to compute significance levels.

2- The use of U2OS cells is not adequate to assess the activation of SREBP-1c by insulin, an hepatocyte cell line would have been preferable. For that reason, the authors have tested whether cholesterol concentration modulates the cleavage which is not relevant with the physiological regulation of SREBP-1c.

We agree with this comment and we moved the original figure 3b to the supplement (supplementary figure 10) and repeated the experiment using the hepatic cell line HepG2, transiently transfected with pFlag-SREBP-1c. The result is very similar, see figure 3 below.

Figure 3: Flag-SREBP1c cleavage reporter vector construction and validation *in vitro* and *in vivo*. (A) The pCDNA3.1-Flag-SREBP-1c vector contains a triple Flag-tag in front of the human *SREBF1* cDNA (Flag-SREBP-1c) and was transiently transfected into HepG2 cells (Figure 3 A). 48 h after transfection, HepG2 cells were incubated in lipid depleted medium (5% LPDS) or 5% LPDS plus the addition of 100 μ M saturated fatty acids (16:0) or unsaturated fatty acids (18:1, 18:2, 16:1 or 20:4, respectively), or the SREBP-1c suppressor 25 hydroxycholesterol (25-HC), and incubated for 16 h. Non-transfected HepG2 cells were used as Flag-negative control (n.t). 2 h before harvest, the protease inhibitor ALLN (N-acetyl-leucinyl-leucinyl-norleucinal) was added. (B) Whole cell extracts were subjected to Western Blot (WB). P- and N-SREBP-1c were detected using anti-Flag antibody. Band intensities were measured using ImageJ, NIH. Relative levels of proteolytically cleaved N-SREBP-1c were calculated as the relative fraction of N-SREBP-1c / P-SREBP-1c signal intensities and are presented in the dot plot, n=6/group.

3- Since SREBP-1c expression is regulated by a feed forward mechanism, the expression of *sreb-1c* (mRNA) must be measured in all experiments.

We have performed the qPCRs of endogenous Sreb1 (mRNA), as requested, in all experiments.

4- the experiments with the adenovirus expressing SREBP-1c full length form *in vivo* are more convincing since the authors nicely show that their construct is activated at refeeding with a carbohydrate diet demonstrating the key role of insulin in this activation. However, once again, the authors should show that the refeeding with diets enriched in uFAs and saturated fats has not modified the response to insulin or insulin secretion in these animals.

We measured plasma glucose and insulin levels as well as insulin signaling in the liver and describe the results in the new version of the manuscript. We found that plasma glucose and

insulin levels were slightly induced by saturated fats (sFA-rich diet) as compared to unsaturated fats (uFA-rich diet). See, supplementary figure 13B, below. Insulin signaling in the liver was largely unchanged between the groups. See, supplementary figure 13E and F, below.

B

E

F

Supplementary Figure 13. Wild type mice were injected intravenously with 2×10^9 PFU Ad-Flag-SREBP-1c reporter vector. **(B)** Four days later the mice were fed for 3 consecutive days with a chow diet enriched with palm oil (saturated fatty acid diet, sFA) or flaxseed oil (unsaturated fatty acid diet, uFA). Blood was drawn and plasma was obtained. Glucose was measured from frozen plasma samples using a glucometer. Plasma insulin levels were determined using a mouse insulin ELISA. $n=5-6$ / group. **(E)** Liver extracts were subjected to WB using the antibodies depicted. **(F)** Relative proportions of the respective phospho-proteins / proteins were calculated and are presented in the dot plot. $n=6$ /group. Outliers are shown as light grey dots. Unpaired T-tests were used to compute significance levels.

Reviewer #2 (Remarks to the Author):

In this study, Peña de la Sancha and colleagues tested the hypothesis that unsaturated fatty acids released by ATGL-mediated lipolysis in adipocytes are important regulators of SREBP cleavage and activation. Their main findings indicate that mice deficient in ATGL in adipocytes have a faster recovery of liver SREBP1 cleavage and activation upon carbohydrate refeeding, such effect that was attributed to a reduced lipolysis and serum unsaturated fatty acid content. Authors perform a series of experiments to prove that adipocyte derived unsaturated fatty acids are in fact modulators of liver SREBP cleavage. Please find below some suggestions to help manuscript convey its message.

n of 3 is unacceptable and inappropriate for statistical analysis. Please increase n.

We thank the reviewer for this important point. As requested by the editor, we have performed power calculations, which showed that several experiments presented in figures 1, 2, and 3 were under-powered (see attached file, "Power Calculations"). To appropriately test our hypothesis that ATGL derived fatty acids control SREBP-1c activation in the liver, we have raised the number of animals used for figures 1A-C 6 h & 9 h, 2A-C 6 h & 9 h, 3D, 3E, 3F, 3G, and supplementary figures 1A & B at 6 h and 9 h, and 5A & B at 6 h and 9 h. We did not increase the replicate numbers for fasted and 3 h refed AAKO and ALKO mice in figures 1 and 2, respectively, because here, no N-SREBP-1c was detectable, which is in line with the literature (J.D. Horton et al., 1998). Moreover, the 12 h timepoints were removed as they are not needed for the understanding of the manuscript (if requested we will move them to the supplement). Instead, we raised the number of replicates to 10 or higher at the 6 h and 9 h timepoints after refeeding (see figures 1A & B and 2A & B, below). Moreover, all additional experiments requested by the reviewers were performed with n numbers of 5 or higher. See figures 1G & H, 2G & H, 3B, 3D & F, 4D, and supplementary Figures 1, 4, 5, 8, 11, 12, 13 and 14.

Figure 1: Adipose tissue ATGL regulates SREBP-1c in the liver. (A) Control and AAKO (adipose specific ATGL deficient) mice were fasted for 12 h overnight and subsequently either sacrificed (Fasted) or refed a HChD (high carbohydrate/low-fat diet) and sacrificed at the time points indicated (Refed). Livers were resected, microsomal membrane fractions (MM) and soluble nuclear extracts (NEX) were prepared and subjected to Western Blot (WB) using an SREBP-1c specific antibody. Ponceau-S-stained membranes are shown as loading controls. Each lane represents a liver extract from one mouse. WBs (horizontally separated at 6 and 9 h) represent separate experiments. Band intensities were measured using ImageJ, NIH. **(B)** Levels of proteolytically cleaved N-SREBP-1c were normalized to

protein loading and are presented in the dot plot. n=3-10 / group. Outliers are shown as light grey dots. Unpaired T-tests were used to compute significance levels.

Figure 2: SREBP-1c processing in mice lacking ATGL in the liver. (A) Control and ALKO (liver specific ATGL deficient) mice were fasted for 12 h overnight and subsequently either sacrificed (Fasted) or refed a HChD (high carbohydrate/low-fat diet) and sacrificed at the time points indicated (Refed). Livers were resected and microsomal membrane fractions (MM) and soluble nuclear extracts (NEX) were prepared and subjected to Western Blot (WB) using a SREBP-1c specific antibody. Ponceau-S stained membranes are shown as loading controls. Each lane represents a liver extract from one mouse. WBs horizontally separated at 6 h and 9 h represent separate experiments. Band intensities were measured using ImageJ, NIH. **(B)** Levels of proteolytically cleaved N-SREBP-1c were normalized to protein loading and are presented in the dot plot. n=3-10 / group. Outliers are shown as light grey dots. Unpaired T-tests were used to compute significance levels.

Please report serum insulin and glucose levels for the fasting-refeeding experiment reported in Figure 1 and 2 and different diets reported in Figure 4C.

We measured serum insulin and glucose levels, and report the results in the new supplementary figures 4A & B, 8A & B, 13A & B and 14B & C, see below:

Supplementary Figure 4. AAKO Plasma glucose and insulin. Control and AAKO (adipose specific ATGL deficient) mice were fasted for 12 h overnight and subsequently, either sacrificed (Fasted) or refeed a HChD (high carbohydrate/low fat diet) and sacrificed at the time points indicated (Refed). Blood was drawn and plasma was obtained. **(A)** Plasma glucose concentrations were measured in frozen samples using a glucometer. **(B)** Plasma insulin levels were determined using a mouse insulin ELISA. $n=12$ / group. Unpaired T-tests were used to compute significance levels.

Supplementary Figure 8. ALKO Plasma glucose and insulin. Control and ALKO (liver specific ATGL deficient) mice were fasted for 12 h overnight and subsequently, either sacrificed (Fasted) or refeed a HChD (high carbohydrate diet), and sacrificed at the time-points indicated (Refed). Blood was drawn and plasma was obtained. **(A)** Glucose concentration was measured from frozen plasma samples using a glucometer. **(B)** Plasma insulin levels were determined using a mouse insulin ELISA. $n=5-12$ / group. Outliers are shown as light grey dots. Unpaired T-tests were used to compute significance levels.

A

B

Supplementary Figure 13. Wild type mice were injected intravenously with 2×10^9 PFU AdFlag-SREBP-1c reporter vector. Four days later the mice were (A) either fasted overnight or fasted and refed a high carbohydrate diet (HChD) overnight, or (B) fed for 3 consecutive days with a chow diet enriched with palm oil (saturated fatty acid diet, sFA) or flaxseed oil (unsaturated fatty acid diet, uFA). Blood was drawn and plasma was obtained. (A) Glucose and insulin plasma concentration of fasted and HChD fed mice or (B) uFA or sFA diet fed mice. Glucose was measured from frozen plasma samples using a glucometer. Plasma insulin levels were determined using a mouse insulin ELISA. $n=5-6$ / group.

B

C

Supplementary figure 14. Plasma insulin and glucose levels. Control and AAKO (adipose specific ATGL deficient) mice were fasted for 12 h overnight and subsequently, either sacrificed (Fasted) or fasted overnight and injected with bovine serum albumin complexed oleic acid (18:1). (B) Plasma glucose levels were determined by glucometer from frozen plasma samples $n=6$ / group (C) Plasma insulin levels were determined by mouse insulin ELISA. $n=5$ / group. Unpaired T-tests were used to compute significance levels.

Is it possible that the Atgl ko mice due to low serum fatty acid have a lower insulin level than controls upon fasting and carbohydrate refeeding?

We measured insulin concentrations in Ad-Flag-SREBP-1c injected control and AAKO animals after fasting. See supplementary Figure 14C (fasting), shown below. Moreover, we measured insulin concentrations in control and AAKO animals 6 h and 9 h after re-feeding a HChD. See, supplementary figure 4B, below. We see slightly higher insulin concentrations in fasted AAKO than in controls, and no appreciable differences in insulin concentrations after re-feeding.

Supplementary figure 14. Plasma insulin and glucose levels. Control and AAKO (adipose specific ATGL deficient) mice were fasted for 12 h overnight and subsequently, either sacrificed (Fasted) or fasted overnight and injected with bovine serum albumin complexed oleic acid (18:1). **(C)** Plasma insulin levels were determined by mouse insulin ELISA. n=5 / group. Unpaired T-tests were used to compute significance levels.

Supplementary Figure 4. AAKO Plasma glucose and insulin. Control and AAKO (adipose specific ATGL deficient) mice were fasted for 12 h overnight and subsequently, either sacrificed (Fasted) or refed a HChD (high carbohydrate/low fat diet) and sacrificed at the time points indicated (Refed). Blood was drawn and plasma was obtained. **(B)** Plasma insulin levels were determined using a mouse insulin ELISA. n=5 / group. Unpaired T-tests were used to compute significance levels.

Figure 3E - If adipose tissue is the main source of unsaturated fatty acids for SREBP1 cleavage inhibition and ATGL has a preference for the hydrolysis of unsaturated fatty acids, short-term feeding (3 days) with diets composed of different fatty acids should not affect serum fatty acid composition and SREBP1 cleavage, unless they promote a strong depletion in unsaturated fatty acids in adipose tissue, which is not the case. Please show serum fatty acid profile for this experiment.

We do see a decrease in some of the measured free unsaturated fatty acid species (NEFAs) (18:2, 18:3 and 20:4, respectively) upon HChD refeeding. Feeding of diets enriched in highly unsaturated fatty acids (mainly 18:3 in flax seed oil) versus saturated fatty acids (mainly 16:0) lead to a decrease in 16:0, a sharp increase in 18:3 and, interestingly, also a moderate decrease in 18:1 and 20:4. See new Supplementary Figure 12, below.

Supplementary Figure 12. Plasma Non-esterified fatty acid (NEFA) profile. Wild type mice were injected intravenously with 2×10^9 PFU AdFlag-SREBP-1c reporter vector. Four days later the mice were (A) either fasted overnight or fasted, and refed a high carbohydrate diet (HChD) overnight, or (B) fed for 3 consecutive days with a chow diet enriched with palm oil (saturated fatty acid diet, sFA) or flaxseed oil (unsaturated fatty acid diet, uFA). Blood was drawn and plasma was obtained. Plasma unsaturated NEFA (uNEFA) or saturated NEFA (sNEFA) levels were measured in (A) fasted or HChD groups, or (B) uFA or sFA diet groups using GC/FID (Gas Chromatography-Flame Ionization Detection), $n=5-6$ /group. Unpaired T-tests were used to compute significance levels.

Figure 4 – is it possible that the injected oleic acid is restoring fasting insulin levels in Atgl ko mice that inhibits the srebp cleavage?

Fasting insulin levels in AAKO are slightly above those of controls. See, supplementary figure 14C, below. However, P-SREBP-1c cleavage is activated by insulin in a process requiring small ribosomal subunit protein S6 (S6) phosphorylation by its kinase, p70S6K ribosomal subunit protein S6 (Owen et al., 2012). We do see upregulated S6 phosphorylation by oleic acid injection, but there is no difference between controls and AAKOs. See, figure 4D & E, below.

Supplementary figure 14. Plasma insulin and glucose levels. Control and AAKO (adipose specific ATGL deficient) mice were fasted for 12 h overnight and subsequently, either sacrificed (Fasted) or fasted overnight and injected with bovine serum albumin complexed oleic acid (18:1). **(C)** Plasma insulin levels were determined by mouse insulin ELISA. n=5 / group. Unpaired T-tests were used to compute significance levels.

Figure 4: SREBP-1c cleavage-activation is suppressed by lipolysis derived uFAs (A) Control and AAKO (adipose specific ATGL deficient) mice were injected intravenously with 2×10^9 PFU AdFlag-SREBP-1c. Four days later, they were either fasted overnight, or fasted overnight, injected with bovine serum albumin complexed oleic acid (18:1) 3 h and 9 h after food withdrawal. All mice were sacrificed in the morning. Livers were resected and microsomal membrane fractions (MM) and soluble nuclear extracts (NEX) prepared and subjected to Western Blot (WB). Band intensities were measured using ImageJ, NIH. **(D)** Liver extracts were subjected to WB using the antibodies depicted. A pool of 3 extracts from 6 h HChD refed control mice was used as positive control (Pool 6 h refed). **(E)** Relative proportions of the respective phospho-proteins / proteins were calculated and are presented in the dot plot. n=6 / group. Unpaired T-tests were used to compute significance levels.

Please explain why you depleted membrane lipids in the experiments reported in Figure 4D and E. Is this physiological?

This treatment is not physiological. We only used this method to ensure that we are able to observe the differential localization of SCAP-GFP triggered by the differential lipid treatments because the SCAP-GFP fusion protein reacts very quickly to changes in intracellular lipid concentrations. Moreover, the same method has helped to assess SREBP and SCAP regulation by fatty acids and sterols in several reports (Hua et al., 1996; Lee et al., 2010; Lee et al., 2008; Ou et al., 2001; Shao et al. 2016).

Considering the redundant role of ChREBP in the regulation of de novo lipogenesis, does this faster recovery of SREBP1 cleavage on atgl ko mice upon refeeding leads to higher rates of liver de novo lipogenesis than controls? Is this mechanism of regulation relevant considering that de novo lipogenesis is markedly decreased upon fasting where

How do you conciliate your hypothesis with the increase in SREBP and de novo lipogenesis seen upon insulin resistance in which lipolysis is markedly elevated in adipose tissue?

In our experiments, we see that adipose-Atgl-KO (AAKO) mice show a faster onset of SREBP-1c cleavage activation and a faster onset of the lipogenic SREBP-1c target genes Fasn and Acaca (Acc-1) after refeeding compared to controls. How physiologically relevant this regulation is in the context of SREBP-1c regulated de-novo lipogenesis, cannot be answered satisfactorily with our limited set of experiments that was designed to clarify if ATGL mediated lipolysis may or may not regulate SREBP-1c in the liver. In addition, in insulin resistance, lipolysis is elevated because the effect of insulin on lipolysis inhibition is blunted. The adipose tissue is “insulin insensitive”. Unexpectedly, as you point out, de-novo lipogenesis is not blunted, however, even elevated. Yao et- al., Nature metabolism 2023, for example suggest that this happens via an alternative mechanism (upregulation of WDR6 in mice). Our own findings do not argue against that. On the contrary, if insulin sensitivity leads to reduced lipolysis, it leads to reduced adipose derived NEFAs in the bloodstream, which would, accordingly, lead to decreased suppression of SREBP-1c cleavage activation in the liver, and, hence, to increased de-novo lipogenesis. These are, however, very theoretical thoughts that clearly go beyond the scope of our current manuscript. It is actually hard to speculate on the role of ATGL derived NEFAs on lipogenesis in diseased state, when we only now propose this mechanism to the scientific community. Nevertheless, we are grateful for your comments, which show that our proposed mechanism may stimulate lively discussions, new experiments and more discussion and hence, strongly stimulate the advance of our fascinating field!

Reviewers' comments:

Reviewer #1 (Remarks to the Author):

Thanks to the authors for adding the results on insulin concentrations and its signaling. However, given the heterogeneity of the samples in the fed state (fig1G), it would have been more judicious to present the fasting state

Dear Reviewer, thank you for your comments.

We present fasting plasma insulin levels of AAKO mice in the Supplementary Fig. 13C. See also below.

Supplementary figure 13.

Insulin signaling analysis of fasted AAKO and control mice is shown in the new Figure 5. Please note that while p-AKT was undetectable during fasting, the quality of the p-AKT antibody was confirmed by the positive control on the right side of the blot.

Figure 5

Reviewer #2 (Remarks to the Author):

Authors have addressed some of my previous concerns, but manuscript still contains several inconsistencies that need to be addressed specially regarding a role of insulin regulation srebp cleavage. In fact, either changes in insulin or insulin signaling could, in all experiments, explain the changes in SREBP cleavage.

Dear Reviewer, thank you for your comments. We have changed the text of the abstract and of the introduction to clarify the role of insulin in transcriptional and post-translational activation of SREBP-1c. See lines 3-5 of the abstract:

“Low levels of unsaturated FAs (uFAs) and high insulin promote its proteolytic activation, yielding N-SREBP-1c that drives fatty acid (FA) biosynthesis.”

See lines 27-28 in the introduction:

“High insulin levels activate both, Srebf1 transcription and cleavage-activation of its protein product P-SREBP-1c.”

Abstract

Lines 9-11- There is a transitory increase in liver SREBP after 6h in adipocyte atgl Ko mice, but not 9 h of refeeding. Therefore, the term hyper-activated is exaggerated. Please describe findings specifically.

This is true and we changed the text accordingly (lines 9-11):

“Indeed, we show that (I) N-SREBP-1c is transiently higher abundant in livers of fasted and re-fed adipose specific Atgl knockout mice than in control livers.”

Lines 14-15- It is important to mention that fatty acids seem to play an inhibitory role on SREBP cleavage mainly upon refeeding, not upon fasting.

We changed the text accordingly (lines 13-15):

“Our findings demonstrate that adipose tissue ATGL derived uFAs attenuate P-SREBP-1c activation in the liver mainly after refeeding.”

Lines 16-19 “This homeostatic regulation...” Please delete this. Too much speculation.

We deleted the passage on “homeostatic regulation”. See, new text (lines 15-16):

“We propose that this ATGL/SREBP 1c axis adds an additional layer of coordination between lipogenesis and lipolysis.”

Results

Figure 1 - There is a transitory increase in liver N-SREBP content after 6h in adipocyte ATGLKO mice, but not 3 h or 9 h of refeeding. Upon fasting, there are no differences in N-SREBP-1c, FASN and ACACAC between genotypes. After refeeding for 3 h, SREBP is still at levels found upon fasting and there are no differences between genotypes, but Srebf, Fasn and Acaca mRNA levels are higher in adipocyte ATGLKO. This indicates that SREBP is not driving this increase in FASN and ACACA mRNA levels found upon 3 h of refeeding, as it requires cleavage for activation.

We agree that N-SREBP-1c is not detected by our western blots (WB) during fasting and 3 h after re-feeding. Nevertheless, it is possible that a small amount of SREBP-1c (which is not detected by WB) is proteolytically activated and drives target gene expression. This can explain why AAKO mice show higher liver *Fasn* (mRNA) and *Acaca* (mRNA) levels. We have now added this explanation also to our “Discussion” section lines 307-314:

“Accordingly, the SREBP-1c target genes Fasn, Acaca and Srebf1 reacted earlier and stronger after re-feeding in AAKO compared to controls (Figure 1C and Supplementary figure 1A). However, we wondered why SREBP 1c target genes reacted before N-SREBP-1c was detected by WB. One possible explanation for this is that small amounts of N-SREBP-1c, which are below WB detection limit, may already drive expression of its target genes. Moreover, Horton et al. were able to detect N SREBP-1c already 3 h after re-feeding using a quite similar experimental setup (<https://www.pnas.org/doi/pdf/10.1073/pnas.95.11.5987>).”

Furthermore, you are showing a reduction in serum fatty acid levels between genotypes upon fasting, but the difference in SREBP occurs only after 6 h of refeeding. At this time point (6h refeeding), serum fatty acids are higher in adipocyte ATGLKO than controls and there are no changes in serum unsaturated fatty acids, and liver fatty acid or TAG contents between genotypes. These findings go against your hypothesis and conclusions.

Experiments *in vitro* have shown that it takes at least 4 h to suppress SREBP-1c cleavage-activation after unsaturated fatty acid treatment. Please see (<https://pubmed.ncbi.nlm.nih.gov/11085986/>). We pasted the relevant figure and the figure legend, below.

Figure 3. Immunoblot analysis (A and B), RNase protection assay (C), and Northern analysis (D) of endogenous SREBPs in HEK-293 cells treated with arachidonate for various times. On day 0, HEK-293 cells were set up as described under “Experimental Procedures.” On day 2, the cells were refed with medium A containing 5% delipidated FCS. On day 3, each dish received fresh medium containing 100 μ M sodium arachidonate in a final concentration of 0.1% BSA. Additions were made in a staggered fashion so that all cells were harvested at the same time. After incubation for the indicated time, the cells were harvested in two groups A and B, from one group, nuclear extract and membrane fractions were prepared. Aliquots of the membranes (50 μ g) and nuclear extracts (50 μ g) were subjected to SDS-PAGE. Immunoblot analysis was carried out with 5 μ g/ml rabbit anti-SREBP-1 IgG (lanes A–F) or anti-SREBP-2 IgG (lanes G–L). The filters were exposed to film for 15 s (membranes) or 5 s (nuclear extracts). P and N denote the precursor and cleaved nuclear forms of SREBP-1 or SREBP-2, respectively. Quantification of bands was done as described in Fig. 1, and the values are expressed relative to the signal intensity of the zero-time value (average of lanes A and F for SREBP-1 and lanes G and L for SREBP-2). C, from the second group, total RNA was isolated and aliquots of total RNA (50 μ g) from pooled dishes were hybridized for 10 min at 68 °C to the ³²P-labeled cRNA probes. Protected fragments were separated by gel electrophoresis and exposed to film for 48 h (SREBP) or 8 h (β -actin) at –80 °C. D, aliquots of total RNA (30 μ g/lane) were subjected to electrophoresis and Northern blot hybridization with the indicated ³²P-labeled probes. The filters were exposed to film for 4 h (SREBP-2) or 0.5 h (β -actin) at –80 °C. The data in panels C and D were quantified as described in Fig. 2.

This indicates that the SREBP-1c cleavage activation system does not respond immediately to exogenous fatty acid levels and may explain why we still see SREBP-1c cleavage-suppression in controls as compared to AAKO, 6 h post re-feeding, even if unsaturated fatty acid levels already returned to “normal” at this timepoint. We have now added this explanation also to our “Discussion” section lines 314-319:

“Another question that remained was why SREBP-1c cleavage activation was suppressed in controls as compared to AAKO 6h after re-feeding (Figure 1A), even though liver uFA levels had already returned to similar levels as in controls (Supplementary figure 2). This may be explained by the relatively slow kinetic of SREBP-1c cleavage-activation in response to exogenous fatty acid levels, which was reported by Hannah et al., in 2001 (<https://pubmed.ncbi.nlm.nih.gov/11085986/>).”

Why adipocyte ATGLKO display higher Srebf mRNA levels at fasting and SREBF, FASN and ACACA mRNA upon 3 h refeeding?

As discussed in lines 310-312 of the manuscript, *“One possible explanation for this is that small amounts of N-SREBP-1c, which are below WB detection limit, may already drive expression of its target genes”.*

In panel G, it is impossible to truly evaluate phospho Akt results. Upon 6 h refeeding, there are 3 adipocyte ATGLKO mice that show levels higher than at least 5 controls, but in the remaining 4 adipocyte ATGLKO mice there is a white blur indicative of issues in protein transference or blotting revelation. Similar issues are found in phospho Akt at 9 h.

We agree that WB quality made it difficult to evaluate AKT phosphorylation. We have therefore, repeated the WBs and the P-AKT/AKT analysis. See Figure 1G and H, respectively. No differences were found in P-AKT/AKT regulation between AAKO and controls at 6 and 9h refeeding.

Figure 1.

The only convincing blotting result is the increase in phospho S6 content in liver upon 6 h refeeding, indicating a major role of insulin promoting SREBP cleavage at this time point, not a reduction in unsaturated fatty acids.

We agree that higher levels of N-SREBP-1c in 6 h re-fed AAKO compared to controls may, at least in part, be mediated by tissue specific insulin signaling. We now acknowledge this in the lines 100-103:

“Collectively, our findings and the cited literature suggest that reduced availability of uFAs and enhanced tissue specific insulin sensitivity lead to increased SREBP 1c cleavage (Figure 1A & B) and upregulated SREBP-1c target gene activation (Figure 1C) in livers of AAKO compared to controls”.

Lines 98-106 – results do not support any role for fatty acids, but a clear role of insulin via S6K/S6 promoting SREBP cleavage at 6 h refeeding. This could be further tested by acutely inhibiting mTORC1 with rapamycin.

As pointed out above, we have changed the text to clearly acknowledge the role of enhanced tissue specific insulin sensitivity in the adipose *Atgl* knockout model in the lines 97-100. Moreover, we again point this out at the end of the discussion section in lines 384-388:

“Our studies fill this gap and demonstrate in vivo the importance of adipose derived uFAs and ATGL for homeostatic control of hepatic lipogenesis. This, however, does not rule out an additional effect caused by

changes in insulin sensitivity. In fact, redundancy in the regulation of a biological important mechanism seems possible.”

Figure 2. The findings that mice bearing liver atgl deletion reproduces the same increase in N-SREBP upon 6 h refeeding shown by mice with adipocyte atgl deletion is puzzling specially because this is not associated with an increase in phospho S6 content as it does upon adipocyte atgl deletion. Again in Figure 2, phospho Akt blots are of very low quality. Why atgl deletion in either hepatocytes or adipocytes results in a similar SREBP phenotype?

We agree that it is unexpected that deletion of Atgl “either in hepatocytes or adipocytes results in a similar SREBP phenotype.” However, literature research suggested that the liver steatosis seen in ALKO mice (*Atgl* liver knockout) may be underlying their enhanced SREBP-1c activation compared to controls. Liver steatosis can lead to ER stress, and, as a result, SREBP-1c is activated via a non-canonical pathway. We have incorporated this line of thought now in the text: lines 144:

“The reason for that may be liver steatosis in the ALKOs” (<https://pubmed.ncbi.nlm.nih.gov/21029304/> & <https://pubmed.ncbi.nlm.nih.gov/30220454/>).

See also discussion section, lines 333 - 338:

“One possible explanation is the liver steatosis of ALKO mice (Figure 2F, Supplementary Figure 7A and 7B) (<https://pubmed.ncbi.nlm.nih.gov/21465509/>). Ferré and Foufelle pointed out a vicious cycle of hepatic steatosis induced ER-stress that activates P-SREBP-1c mediated lipogenesis, which is independent of insulin signaling. Later on, the group of Michael Karin showed that ER-Stress in steatotic livers activates SREBP-cleavage via Caspase-2 and S1P, in a SCAP independent manner” (<https://pubmed.ncbi.nlm.nih.gov/21029304/> & <https://pubmed.ncbi.nlm.nih.gov/30220454/>).

As for the p-AKT and AKT blots we agree on the poor quality. We have, therefore, repeated these blots, see Figures 2G&H.

Figure 2.

In Figure 3, the inhibitory effect on SREBP cleavage of diet enriched with unsaturated fatty acids occurred along a reduction in serum insulin levels, which complicates the interpretation. This insulin result should be in Figure 3, not supplemental. Again, in Figure 13 supplemental, phospho Akt blots are of very low quality, which precludes an evaluation of insulin signaling.

To better acknowledge the lower serum insulin levels in unsaturated fatty acid rich diet fed animals we changed the text. Lines 214-215:

“Plasma glucose showed no differences while plasma insulin levels were reduced by uFA-rich diet as compared to sFA-rich diet (Figure 4I)”.

Moreover, we have moved insulin measurement plots as well as tissue specific insulin signaling western blots from the supplement to the main figure. As a result, the previous Figure 3 is now split up into Figure 3 (*in vitro* experiments) and Figure 4 (*in vivo* experiments). As for the p-AKT/AKT western blots, we have repeated these (Figure 4 D&E as well as I&J).

Figure 4:

In Figure 4, higher SREBP cleavage occurred in association with higher serum insulin levels in adipocyte atgl ko mice and, 18:1 administration reduced both SREBP cleavage and serum insulin levels. Again, changes in serum insulin could explain all the changes in SREBP.

We apologize for a mistake in the previous Figure 4, where we showed that plasma insulin was higher in AAKO mice during fasting. This was due to the fact that we copied the data for other samples erroneously. After rectification, we corrected the data, which now show that AAKO mice have significantly lower plasma insulin levels than controls (Supplementary Figure 13C).

Supplementary figure 13

This finding is also in agreement with the literature (<https://pubmed.ncbi.nlm.nih.gov/26196542/>). Therefore, we do not think, at least in this experiment, that the upregulation of N-SREBP-1c in livers of fasted AAKO as compared to controls is due to an insulin effect.

In addition, in Figure 5, we see that after 18:1 uFA infusion, despite a S6 Ser 240/244 increase in both groups (Figure 5D&E), N-SREBP-1c was downregulated in AAKO mice and closely matched N-SREBP-1c levels of controls (Figure 5B&C).

Figure 5.

** See the Nature Portfolio author and referees' website at www.nature.com/authors for information about policies, services and author benefits

Communications Biology is committed to improving transparency in authorship. As part of our efforts in this direction, we are now requesting that all authors identified as 'corresponding author' create and link their Open Researcher and Contributor Identifier (ORCID) with their account on the Manuscript Tracking System prior to acceptance. ORCID helps the scientific community achieve unambiguous attribution of all scholarly contributions. You can create and link your ORCID from the home page of the Manuscript Tracking System by clicking on 'Modify my Springer Nature account' and following the instructions in the link below. Please also inform all co-authors that they can add their ORCIDs to their accounts and that they must do so prior to acceptance.

If you experience problems in linking your ORCID, please contact the Platform Support Helpdesk.

This email has been sent through the Springer Nature Tracking System NY-610A-NPG&MTS

Confidentiality Statement: